# Oncogenic KRAS is dependent upon an EFR3A-PI4KA signaling axis for potent tumorigenic activity

Hema Adhikari[1], Walaa E. Kattan[2,3], Shivesh Kumar [4], Pei Zhou [4], John F. Hancock[2,3 ✉] &
Christopher M. Counter [1 ✉]

The *HRAS*, *NRAS*, and *KRAS* genes are collectively mutated in a fifth of all human cancers. These mutations render RAS GTP-bound and active, constitutively binding effector proteins to promote signaling conducive to tumorigenic growth. To further elucidate how RAS oncoproteins signal, we mined RAS interactomes for potential vulnerabilities. Here we identify EFR3A, an adapter protein for the phosphatidylinositol kinase PI4KA, to preferentially bind oncogenic KRAS. Disrupting EFR3A or PI4KA reduces phosphatidylinositol-4-phosphate, phosphatidylserine, and KRAS levels at the plasma membrane, as well as oncogenic signaling and tumorigenesis, phenotypes rescued by tethering PI4KA to the plasma membrane. Finally, we show that a selective PI4KA inhibitor augments the antineoplastic activity of the KRAS[G12C] inhibitor sotorasib, suggesting a clinical path to exploit this pathway. In sum, we have discovered a distinct KRAS signaling axis with actionable therapeutic potential for the treatment of KRAS-mutant cancers.

[1] Department of Pharmacology & Cancer Biology, Duke University, Durham, NC, USA. [2] Integrative Biology & Pharmacology, McGovern Medical School, University of Texas Health Science Center, Houston, TX, USA. [3] The University of Texas MD Anderson Cancer Center UTHealth Graduate School of Biomedical Sciences at Houston, Houston, TX, USA. [4] Department of Biochemistry, Duke University, Durham, NC, USA. ✉email: john.f.hancock@uth.tmc.edu; count004@mc.duke.edu

The three RAS genes HRAS, NRAS, and KRAS are collectively mutated in a fifth of human cancers[1]. These mutations primarily alter residues G12, G13, or Q61[2], which leave RAS in a constitutively active and oncogenic GTP-bound state[3]. Oncogenic RAS recruits effector proteins, the most studied being RAFs, PI3Ks, and RalGEFs[3], to the plasma membrane where RAS resides to propagate oncogenic signaling[3]. Excitingly, small molecules specifically inhibiting the G12C-mutant variant of oncogenic KRAS have been shown to have therapeutic potential in human clinical trials[4,5], with one, sotorasib, now approved for the treatment of lung cancer. Nevertheless, consistent with being a targeted therapy[6,7], early indications suggest resistance arises through upregulating the EGFR pathway or by increasing RAS oncoprotein levels[8,9]. As such, there is a great clinical need to identify components of RAS signaling that could be leveraged to either augment the antineoplastic activity of, or combat resistance to these and future RAS inhibitors. As RAS is a signaling protein, the most logical source of such potential vulnerabilities are proteins in the immediate vicinity of the oncoprotein itself.

As a signaling protein, RAS is subjected to multiple levels of regulation[10]. One such level is the spatial-temporal control of the protein at membranes[11,12]. Each RAS isoform is prenylated at the C-terminus to foster membrane association[10,13,14]. Careful monitoring of the subcellular localization of RAS revealed that the protein cycles back and forth between internal membranes and the plasma membrane[15–17], although the latter is considered to be the primary site of signaling[18,19]. While at the plasma membrane, activated RAS forms nanoclusters of six to seven RAS proteins[11,19], which facilitate signaling by favoring the formation of oncoprotein signaling complexes[19–22]. Not surprisingly, the content of the plasma membrane affects the occupancy and nanoclustering of RAS in this locale, providing yet another layer of regulation[23]. Indeed, we previously found that a phosphatidylinositol-4-phosphate (PI(4)P) concentration gradient promotes and maintains KRAS localization and nanoclustering at the plasma membrane through an exchange with phosphatidylserine (PS) at contact sites between this membrane and the endoplasmic reticulum[24]. As such, it stands to reason that KRAS-associating proteins that affect PI(4)P metabolism may offer a unique way to not only regulate, but perhaps even to directly target the spatial-temporal regulation of this oncoprotein.

PI(4)P is typically generated by phosphorylating the D4 position in the inositol headgroup of phosphatidylinositol (PI) by PI(4)P kinases, one of which is PI4KA[25,26]. PI4KA is recruited to plasma membrane by the adapter protein EFR3[27,28], which has two isoforms, EFR3A and EFR3B[29]. These two proteins are evolutionarily conserved from mammals to yeast, and share 62% sequence identity[29]. EFR3A/B contain an N-terminal cysteine rich region that encodes palmitoylation sites for plasma membrane anchoring, followed by armadillo-like repeats that mediate protein-protein interactions[29]. EFR3A/B form a complex with PI4KA and two other proteins, FAM126A and TTC7A or TTC7B, which serve as scaffolding proteins stabilizing PI4KA at the plasma membrane[30–32]. While mutations in EFR3A have been associated with autism[33] and neurogenesis[34], no direct relationship between this protein and human cancers had been established. Although as adapter proteins EFR3A/B do not easily lend themselves to therapeutic intervention, the recruitment of PI4KA does. Lipid kinases can be inhibited with small molecules[35,36], with a PI3K inhibitor approved for cancer treatment[37]. As such, a connection between PI(4)P and oncogenic KRAS could potentially be a druggable avenue to inhibit the regulation of the oncoprotein itself.

Critical to unmasking components involved in the spatial-temporal regulation of oncogenic RAS is identifying proteins associating with the oncoprotein, particularly those linked to the plasma membrane. While a number of specific protein-protein interactions had been documented over the years, which has greatly informed current understanding of RAS signaling[3,38], a comprehensive characterization of RAS interactomes was lacking. To this end, BirA-mediated proximity labeling, which can identify proteins within 10 nm distance of a targeted protein[39,40], had been used to define the interactome of oncogenic KRAS[41]. We[42] and others[41,43,44] similarly used this approach to define the interactomes of all three RAS oncoproteins. Not only did we identify 65 previously documented interacting proteins, but by comparing the interactomes of each RAS isoform, we identified the PI kinase PI5K1A as preferentially binding to and mediating oncogenic KRAS signaling and cellular proliferation[42]. One advantage of BirA-proximity labeling over biochemical purification of protein complexes is that transient and/or weak interactions can be captured, which are characteristic of regulatory proteins[40]. We thus sought to identify pristine interactions with RAS oncoproteins essential for oncogenesis. To this end, we report here that oncogenic RAS associates with EFR3A. Focusing on KRAS, as it is the most commonly mutated of the three RAS genes in human cancers[1], we further show that that loss of EFR3A/B or PI4KA reduces the levels of PI(4)P and PS, as well as KRAS localization and nanoclustering at the plasma membrane. This is associated with reduced oncogenic KRAS signaling, transformation, and tumorigenesis. Importantly, epistatic analysis confirms these results, as tethering either PI4KA or RAS to the plasma membrane rescues these effects. Thus, EFR3A appears to form a unique positive-feedback circuit whereby activated RAS binds EFR3A, which in turn promotes signaling by the oncoprotein. Such a circuit opens up the possibility of pharmacologically augmenting the antineoplastic activity of RAS inhibitors with PI4KA inhibitors, and in agreement, we demonstrate that a selective PI4KA inhibitor is synergistic with the clinical KRAS$^{G12C}$ inhibitor sotorasib.

## Results

**Identification of EFR3A from a RAS interactome screen**. We previously performed BirA-mediated proximity labeling with each of the three oncogenic RAS isoforms (HRAS$^{G12V}$, NRAS$^{G12V}$, KRAS$^{G12V}$), identifying 474 'interactome' proteins. Each of these proteins were then analyzed for their effect on RAS oncogenesis in a loss-of-function CRISPR/Cas9 screen using a panel of isogenic cell lines transformed by a different oncogenic RAS isoform. Comparing these two datasets uncovered PIP5K1A to specifically bind to and promote KRAS oncogenesis, validating the approach[42]. As noted above, a distinct advantage of BirA-mediated proximity labeling over conventional pull-down and other proteomic approaches is the identification of weak and/or transient interactions[39,40], which are often characteristic of regulatory proteins[40]. Given this, we mined the two aforementioned datasets for interactome components that are particularly critical for oncogenesis, which identified EFR3A as the top candidate. In more detail, EFR3A was biotinylated by all three BirA-HRAS$^{G12V}$, BirA-NRAS$^{G12V}$, and BirA-KRAS$^{G12V}$ fusion proteins, roughly at par with the known RAS effectors BRAF and CRAF (Supplementary Fig. 1a–c). Encouragingly, mining the BirA-KRAS$^{G12V}$ dataset also revealed that EFR3B and FAM126A were biotinylated at a low level, as was PI4KA, although as only one peptide was recovered from the latter, PI4KA was excluded from our previous analysis[42]. EFR3A, PI4KA, FAM126A, and TTC7B were also variably biotinylated in most[43,44], but not all[41], other reported proximity labeling experiments using BirA-RAS oncoproteins. Furthermore, EFR3A sgRNAs were consistently the most negatively enriched of all targeted interactome candidates in cells transformed by oncogenic RAS in our dataset (Fig. 1a)[42],

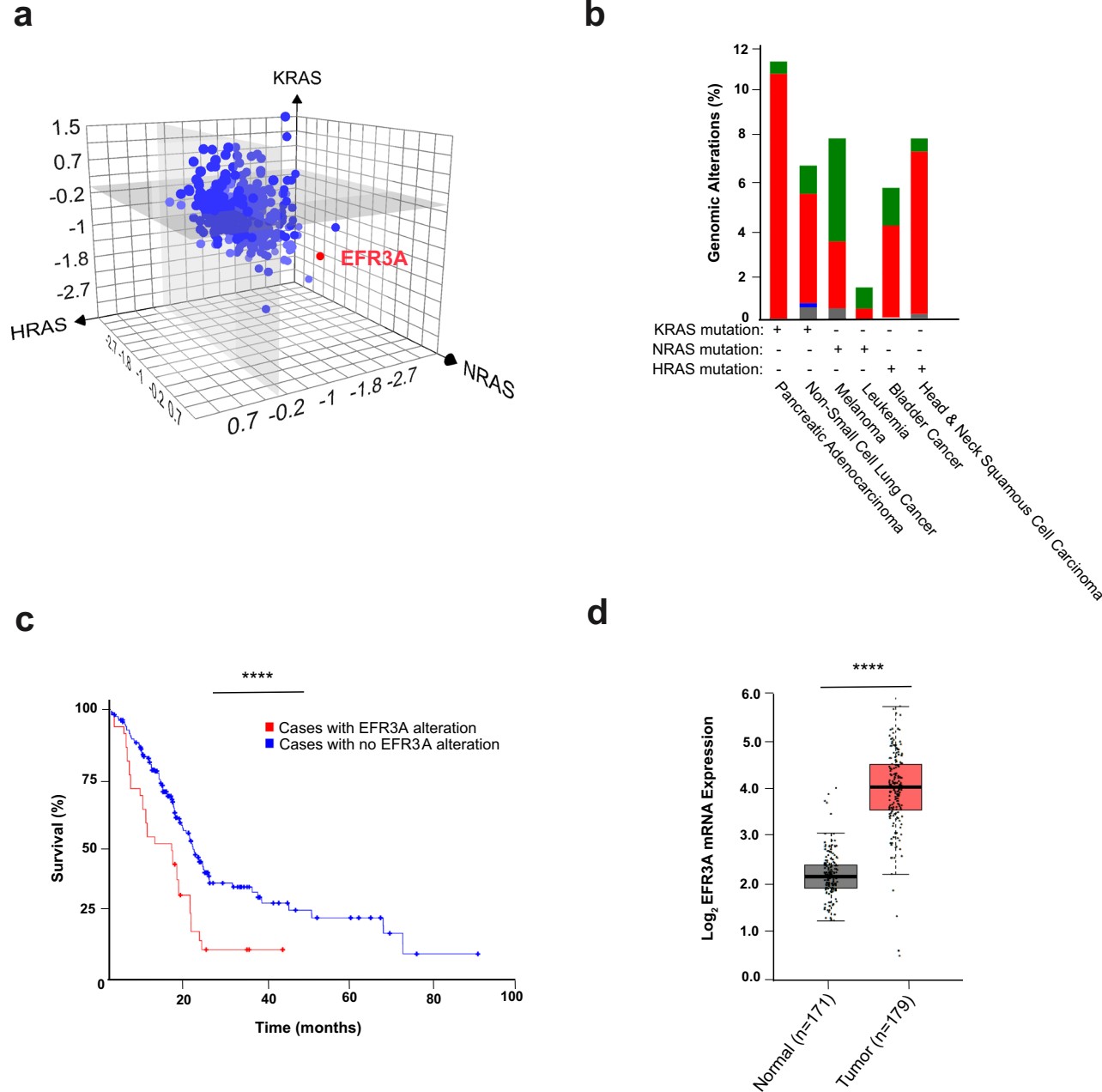

**Fig. 1 Identification of EFR3A as a potential vulnerability in KRAS-mutant pancreatic cancer. a** Mean $\log_2$ fold-enrichment of the 5 sgRNAs targeting each of the 474 genes encoding proteins of the RAS interactomes (determined by BirA-mediated proximity labeling) in the isogenic background of HEK-HT cells transformed by oncogenic (G12V) *HRAS* (*x*-axis), *KRAS* (*y*-axis), or *NRAS* (*z*-axis). **b** The frequency (%) that the *EFR3A* gene exhibits amplification and copy number variation (red bar), a point mutation (green bar), deletion (blue bar), or multiple alterations (gray bar) in 6 different RAS-mutant cancers from the cBiportal PanCan TCGA dataset. Only cancers with ≥2% genomic alterations are shown. **c** A Kaplan–Meier plot of overall survival of pancreatic adenocarcinoma cases with (*n* = 42) versus without (*n* = 327) a genomic alteration in *EFR3A* (*p* = 2.034e-4), as determined from 5 pancreatic cancer datasets (ICGC, Nature 2012; TCGA PanCancer Atlas; UTSW Pancreatic Cancer; TCGA Firehose Legacy; QCMG, and Nature 2016), *p* = 0.01504. **d** A boxplot of the $\log_2$ mRNA expression of *EFR3A*, normal max (3.1), min (1.3), center (2.2); tumor max (4.5), min (0.3), center (4.2) in 179 pancreatic adenocarcinoma versus 171 normal tissues based on GEPIA analysis of the PAAD dataset. Significance values calculated by one-sided logrank test for (**c**) and one-way ANOVA analysis for (**d**): ****$p < 0.0001$. Percentiles in the box plots pre-set at 50% (higher) and 50% (lower).

and also negatively enriched in other similar datasets[44]. To put this into context, the average negative enrichment score for all five *EFR3A* sgRNAs in cells transformed by any RAS oncoprotein was almost twice that of *BRAF* sgRNAs, which again target a known RAS effector[3] (Supplementary Fig. 1d). We note that *CRAF* sgRNA depletion was not as prominent as *BRAF* sgRNA depletion (Supplementary Fig. 1d), as seen on other screens[45]. Given these findings we explored EFR3A further.

**EFR3A is upregulated in pancreatic cancer.** To explore *EFR3A* in the context of human cancers we surveyed the cBioportal cancer genomics datasets for genomic alterations in the *EFR3A* gene. Across all human cancers (*n* = 10,967 samples) there was a general bias towards *EFR3A* amplification, with ovarian, eso-phageal, breast, pancreatic, and liver cancers all exhibiting over 10% gene amplification (Supplementary Fig. 2a). Comparing *EFR3A* gene amplification in examples of cancers driven by a

specific RAS isoform, we find *EFR3A* gene amplification is highest in pancreatic adenocarcinoma (Fig. 1b). Given that pancreatic adenocarcinoma has both the highest frequency of *KRAS* mutations and *EFR3A* amplification, we focused on this cancer. We find that pancreatic adenocarcinoma patients with an *EFR3A* genomic alteration (almost exclusively gene amplification, Fig. 1b) have a 26% reduction in overall median survival compared to patients without an *EFR3A* genomic alteration (i.e. 15.3 versus 20.8 months, Fig. 1c). This also manifested as a 53% reduction in disease-free progression in the same cohort (i.e. 9.6 versus 20.5 months, Supplementary Fig. 2b). At the gene expression level, GEPIA analysis of the PAAD cancer dataset revealed a significant 95% increase in relative *EFR3A* mRNA level in pancreatic adenocarcinoma tumors ($n = 179$) compared to matched normal ($n = 171$) tissues (Fig. 1d). Although a limited number of samples, differentiating *KRAS* mutation-positive ($n = 64$) from *KRAS* mutation-negative ($n = 115$) pancreatic adenocarcinoma samples revealed higher *EFR3A* expression in the former (Supplementary Fig. 2c). *EFR3A* gene amplification also tracked with *KRAS*-mutant pancreatic adenocarcinoma samples (Supplementary Fig. 2d). Finally, comparing the survival of pancreatic adenocarcinoma patients with high ($n = 44$) versus low ($n = 46$) *EFR3A* mRNA levels from the TGCA database revealed a 29% reduction in overall median survival of the former (Supplementary Fig. 2e). Pancreatic adenocarcinoma appears to be unique among RAS-driven cancers in this regard, as high *EFR3A* expression was not observed in either other KRAS-driven cancers examined, or cancers driven by mutations in other RAS-isoforms (Supplementary Fig. 3a). Given these findings, we explored whether expression of genes encoding the other components of the EFR3A signaling complex was increased in pancreatic adenocarcinoma, finding elevated mRNA levels for *EFR3B* (Supplementary Fig. 3b), *PI4KA* (Supplementary Fig. 3c), *FAM126A* (Supplementary Fig. 3d), and *TTC7A* (Supplementary Fig. 3e). In sum, the EFR3A complex appears to be preferentially upregulated in *KRAS*-mutant pancreatic adenocarcinoma, which is associated with a worse clinical outcome.

**Loss of EFR3A inhibits oncogenic KRAS signaling and transformation.** Given the labeling of multiple components of the EFR3-TTC7-FAM126-PI4KA complex by BirA-KRAS[G12V], the negative enrichment of *EFR3A* sgRNA in oncogenic KRAS-transformed cells, and the above correlation between increased *EFR3A* gene amplification and expression and pancreatic adenocarcinoma, we sought to experimentally test whether inhibiting EFR3A affects KRAS signaling and oncogenesis. To this end, we stably infected KRAS[G12V]-transformed HEK-HT cells, which depend upon KRAS[G12V] for tumorigenesis[46], with a lentivirus encoding Cas9 in the absence (vector) or presence of an sgRNA targeting *EFR3A* (sgEFR3A), and confirmed by immunoblot reduction in detectable endogenous EFR3A protein in the latter cells (Fig. 2a). To assess the effect on oncogenic RAS signaling, we assayed the levels of phosphorylated ERK1/2 (P-ERK) and AKT (P-AKT) by immunoblot as a measure of oncogenic KRAS activation of the downstream MAPK and PI3K/AKT effector pathways[3]. This revealed a prominent reduction in oncogenic KRAS signaling upon the loss of EFR3A (Fig. 2a). These results were corroborated with a second independent sgRNA (sgEFR3A-2) targeting *EFR3A* (Fig. 2a). To evaluate the effect of EFR3A loss on KRAS oncogenesis, we measured colony formation, a known two-dimensional (2D)-transformed growth phenotype of oncogenic KRAS[46], in the same cells, finding a reduction in this transformed phenotype upon the loss of EFR3A expression. We validated these results with an sgRNA targeting the *EFR3B* gene (sgEFR3B), and further show that the combination of sgEFR3A

and sgEFR3B had the greatest reduction on colony formation (Fig. 2b, c). Finally, to evaluate the effect of EFR3A loss on a more aggressive transformed phenotype, we test and found that two independent sgEFR3As, sgEFR3B, or both sgEFR3A and sgEFR3B together all reduced three-dimensional (3D)-transformed growth, as assessed by colony formation in soft agar, of HEK-HT cells transformed with a codon-optimized KRAS[G12V] (Fig. 2d). In sum, the loss of EFR3A inhibits oncogenic KRAS signaling and transformation.

**Loss of EFR3A inhibits pancreatic tumorigenesis.** To evaluate the role of endogenous EFR3A in a cancer driven by an oncogenic mutation in the native *KRAS* gene, we targeted *EFR3A*, *EFR3B*, or both genes by CRISPR/Cas9-mediated gene inactivation as above in the *KRAS*-mutant human pancreatic adenocarcinoma cell line HPAF-II[47]. Immunoblot analysis confirmed the appropriate loss of endogenous protein expression of the targeted genes as well as a corresponding reduction in P-ERK and P-AKT levels (Fig. 3a). To independently validate these results, we repeated the experiment in the *KRAS*-mutant human pancreatic adenocarcinoma cell lines PANC-1, AsPC-1, and CFPAC-1[47]. Again, targeting *EFR3A*, *EFR3B*, or both genes inhibited oncogenic KRAS signaling, with the loss of both isoforms having the greatest effect (Fig. 3a). To evaluate the effect of ERF3A loss on the pancreatic adenocarcinoma cell growth, these 20 cell lines were seeded at low density, after which the number of viable cells was determined over time using the CellTiter Glo reagent. In all cases, both *EFR3A* sgRNAs reduced cell counts compared to vector controls in all four cell lines, while the effect of *EFR3B* sgRNA was more variable, with the combined targeting of both isoforms being as or more potent than *EFR3A* sgRNA alone (Fig. 3b–e). As above, we validated these results in 3D-transformed growth, again finding that loss of *EFR3A*, *EFR3B*, or both reduced colony formation of all four pancreatic adenocarcinoma cells in soft agar (Supplementary Fig. 4a). Next, we evaluated the effect of EFR3A/B loss on the KRAS signaling and transformation in a rare KRAS mutation-negative human pancreatic adenocarcinoma cell line, finding that *EFR3A* and/or *EFR3B* sgRNA had little to no effect on the levels of P-ERK or P-AKT (Supplementary Fig. 4b), 2D- (Supplementary Fig. 4c, d), and 3D- (Supplementary Fig. 4a) transformed growth. To evaluate whether EFR3A was similarly required for *KRAS*-mutant pancreatic tumor growth, the aforementioned HPAF-II cells, in which the expression of *EFR3A*, *EFR3B*, or both genes was verified to be reduced (Fig. 3f), were injected subcutaneously into immunocompromised mice. Tumor growth was then compared over time to that of similarly injected vector control cells. Inactivating these genes individually or together potently inhibited tumor growth, as evidence by a 4.5-fold reduction at the experimental end-point compared to vector control cells (Fig. 3f and Supplementary Fig. 4e). We show that this reduction in tumor growth was, as in 2D- and 3D-transformed growth assays, associated with a reduction in oncogenic KRAS signaling, namely we observed a reduction in P-ERK and P-AKT levels in tumors in which *EFR3A*, *EFR3B*, or both genes were inactivated (Fig. 3g). In sum, loss of EFR3A (and/or EFR3B) expression inhibits oncogenic signaling in human *KRAS*-mutant pancreatic adenocarcinoma cells, which manifests as a reduction in transformed and tumorigenic growth.

**Cellular EFR3A preferentially binds to active KRAS.** To elucidate the mechanism underpinning the effects of EFR3A on KRAS oncogenesis, we examined the association of EFR3A with oncogenic KRAS. We first validated the association of the two proteins by ectopically co-expressing FLAG epitope-tagged EFR3A (FLAG-EFR3A) with myc epitope-tagged BirA-KRAS[G12V]

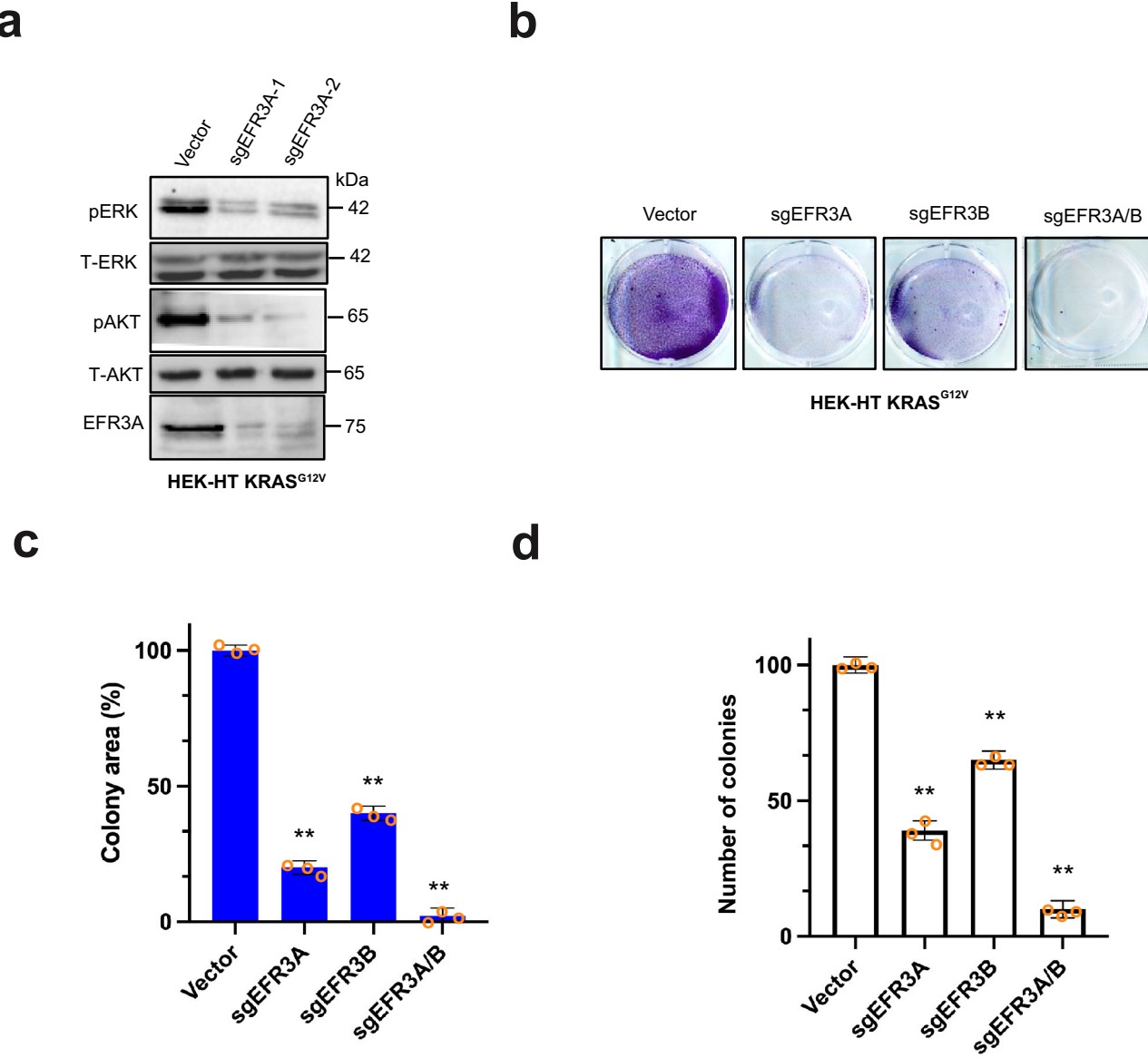

**Fig. 2 EFR3A loss inhibits oncogenic RAS signaling and transformed growth in isogenic cells. a–d** Analysis of the (**a**) levels of phosphorylated (P) and/or total (T) ERK, AKT, EFR3A, and actin (as a loading control), as assessed by immunoblot, **b** an example of colony formation, as assessed by crystal violet staining, which is (**c**) plotted as the mean ± SD total colony area, and (**d**) the mean ± SD number of colonies growing in soft agar of human HEK-HT cells transformed by oncogenic KRAS$^{G12V}$ (codon-optimized in **d**) engineered to stably express Cas9 and a vector encoding no sgRNA (vector) or the indicated sgRNAs. Representative of 3 biological replicates tested in 3 independent experiments. Replicate experiments and full-length gels are provided in Supplementary Fig. 11. Significance values calculated by one-way ANOVA test for (**c**, **d**): **$p < 0.01$. Specific $p$ values for (**c**) are 0.0078 (*sgEFR3A*), 0.0045 (*sgEFR3B*), and 0.0016 (*sgEFR3A/B*), and (**d**) are 0.0059 (*sgEFR3A*), 0.0083 (*sgEFR3B*), and 0.0026 (*sgEFR3A/B*).

(myc-BirA-KRAS$^{G12V}$) in human 293T cells. After treating these cells with biotin, proteins proximity-labeled by myc-BirA-KRAS$^{G12V}$ were affinity purified with streptavidin and immunoblotted with FLAG- and myc-specific antibodies. FLAG-EFR3A was captured by streptavidin, suggesting that EFR3A lies within the immediate proximity (10 nm) of KRAS$^{G12V}$ (Fig. 4a, b), a physiological distance that mediates protein-protein interactions[39]. To determine if this interaction was dependent on the activation state of KRAS, we repeated this experiment, except we compared the biotin labeling of FLAG-EFR3A by myc-BirA-KRAS with an oncogenic (G12V) mutation, with an inactivating (S17N) mutation[48], or without a mutation (wild-type, WT). We confirmed the appropriate activation status of the three myc-BirA-KRAS proteins by affinity capturing GTP-bound RAS with the _R_as _B_inding _D_omain (RBD) of the effector

protein RAF1 (CRAF)[49] fused to GST. Immunoblot analysis to detect the myc epitope-tag revealed a stepwise increase in amount of active myc-BirA-KRAS captured, from none (S17N), to low (WT), to very high (G12V) (Supplementary Fig. 5a). We next compared the amount of FLAG-EFR3A biotinylated by each of these versions of myc-BirA-KRAS as above, which revealed labeling primarily only by the G12V mutant (Fig. 4c, d), consistent with EFR3A associating with active KRAS in cells.

**EFR3A directly binds to active KRAS through its C-terminus.**
To determine if the association between EFR3A and KRAS is direct, we performed in vitro co-immunoprecipitation assays with recombinant GST-EFR3A protein purified by affinity capture with a glutathione Sepharose 4B resin and recombinant His$_{10}$-KRAS$^{G12V}$

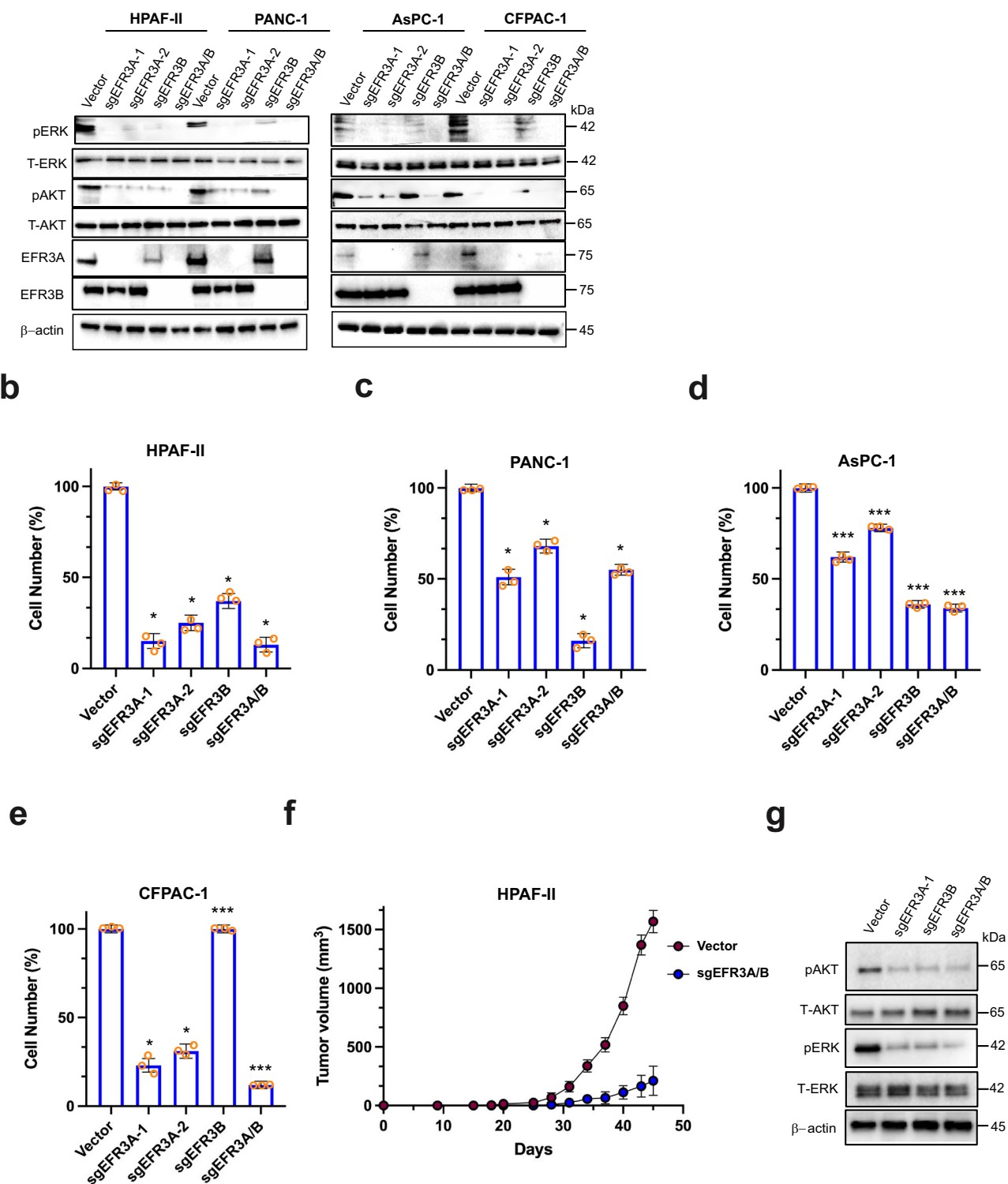

protein purified by affinity capture with a Ni-NTA resin. GST-EFR3A was then incubated with $His_{10}$-$KRAS^{G12V}$, after which $His_{10}$-$KRAS^{G12V}$ was immunoprecipitated with a KRAS-specific antibody and immunoblotted with EFR3A- or KRAS-specific antibodies. This revealed that $His_{10}$-$KRAS^{G12V}$ co-immunoprecipitated with GST-EFR3A (Fig. 4e, f). This interaction was corroborated in the reverse direction. Namely, similarly purified recombinant GST

alone as a negative control or the above GST-EFR3A were incubated with the above $His_{10}$-$KRAS^{G12V}$ protein. GST or GST-EFR3A protein was then immunoprecipitated with a GST-specific antibody followed by immunoblot with GST- or KRAS-specific antibodies. This revealed that $His_{10}$-$KRAS^{G12V}$ co-immunoprecipitated with GST-EFR3A, but not with GST (Supplementary Fig. 5b). To determine if this interaction depended upon the activation status of

**Fig. 3 EFR3A loss inhibits pancreatic adenocarcinoma transformed and tumorigenic growth. a–e** Analysis of the (**a**) levels of phosphorylated (P) and/or total (T)- ERK, AKT, EFR3A, EFR3B, and actin (as a loading control), as assessed by immunoblot, and (**b–e**) the mean ± SD cell number (compared to vector control) over time, as assessed by the Titer Glo assay, of the indicated human *KRAS* mutation-positive pancreatic adenocarcinoma cell lines engineered to stably express Cas9 and a vector encoding no sgRNA (vector) or the indicated sgRNAs. **f** Mean ± SD tumor volume (mm$^3$) versus time of HPAF-II cells engineered to stably express Cas9 and a vector encoding no sgRNA (vector) or the indicated sgRNAs after injected into mice. **g** Levels of (P) and/or (T) ERK, AKT, and actin (as a control), as assessed by immunoblot, in the indicated tumors from panel f. **a**: Representative of 3 biological replicates. **b–e** Representative of 3 biologic replicates assayed in triplicate. **f** The two cell lines were each injected into 5 mice. **g** Representative of 2 biological replicates. Replicate experiments and full-length gels are provided in Supplementary Fig. 11. Significance values calculated by one-way ANOVA test for (**b–f**): *$p < 0.05$ and ***$p < 0.001$. Specific *p* values for (**b**) are 0.0324 (*sgEFR3A-1*), 0.0227 (*sgEFR3A-2*), 0.0428 (*sgEFR3B*), and 0.0487 (*sgEFR3A/B*), (**c**) are 0.0241 (*sgEFR3A-1*), 0.0472 (*sgEFR3A-2*), 0.0281 (*sgEFR3B*), and 0.0307 (*sgEFR3A/B*), **d** are 0.00024 (*sgEFR3A-1*), 0.00062 (*sgEFR3A-2*), 0.00045 (*sgEFR3B*), and 0.00031 (*sgEFR3A/B*), **e** are 0.046 (*sgEFR3A-1*), 0.0381 (*sgEFR3A-2*), 0.00053 (*sgEFR3B*), and 0.00034 (*sgEFR3A/B*), and (**f**) is **$p < 0.01$.

KRAS, we repeated this experiment, except using purified recombinant wild-type (WT) or oncogenic (G12V) His$_{10}$-KRAS incubated with buffer, GDP, or non-hydrolysable GTP (GTPγS). As above, we confirmed the activation status in all six settings by RBD-affinity capture, which led to the predictable outcome, namely that oncogenic His$_{10}$-KRAS$^{G12V}$ was more active than His$_{10}$-KRAS$^{WT}$ in all conditions, and loading with GTPγS was more active than GDP or buffer with both proteins (Fig. 4g). These six versions of His$_{10}$-KRAS were incubated with GST-EFR3A and again His$_{10}$-KRAS$^{G12V}$ or His$_{10}$-KRAS$^{WT}$ was immunoprecipitated with a KRAS-specific antibody and immunoblotted with EFR3A- or KRAS-specific antibodies. The results mirrored the activation status of the KRAS proteins, and ranged from GTPγS-loaded His$_{10}$-KRAS$^{G12V}$ capturing the most GST-EFR3A to buffer or GDP-loaded His$_{10}$-KRAS$^{WT}$ capturing the least (Fig. 4g, h). Thus, recombinant GST-EFR3A preferentially co-immunoprecipitates with GTP-bound or oncogenic recombinant His$_{10}$-KRAS.

To validate these findings using a different detection platform, we repeated the analysis by far-western dot blot, again using purified recombinant proteins (Supplementary Fig. 5d, e). GST as a negative control or GST-EFR3A was affixed to membranes, incubated with His$_{10}$-KRAS$^{G12V}$, and then immunoblotted with a KRAS-specific antibody. Levels of GST, GST-EFR3A, and His$_{10}$-KRAS$^{G12V}$ protein were validated by affixing these proteins to membranes and immunoblotting for their presence. We find that His$_{10}$-KRAS$^{G12V}$ directly bound to GST-EFR3A, but not GST (Fig. 4i, j). To address whether this association was dependent upon the oncogenic status of KRAS, we repeated the study at three concentrations using the above wild-type (WT) or oncogenic (G12V) recombinant His$_{10}$-KRAS proteins with recombinant GST-EFR3A protein. We find that only His$_{10}$-KRAS$^{G12V}$ bound GST-EFR3A (Fig. 4k, l). Thus, far western analysis confirms a direct interaction of recombinant GST-EFR3A with recombinant oncogenic His$_{10}$-KRAS.

To validate these results using a biophysical approach, we examined the interaction between GST-EFR3A and His$_{10}$-KRAS$^{G12V}$ by size-exclusion chromatography (SEC). Recombinantly purified GST-EFR3A and His$_{10}$-KRAS$^{G12V}$ were profiled using the Superose® 6 increase 10/300GL column with peak fractions assayed with EFR3A- or KRAS-specific antibodies to identify the desired proteins. We found that GST-EFR3A eluted as a monodisperse peak at 15.48 ml (Fig. 4m) as detected by immunoblot (Fig. 4n), and His$_{10}$-KRAS$^{G12V}$ eluted as a monodisperse peak at 18.49 ml (Fig. 3o) as detected by immunoblot (Fig. 4p). Consistent with the expected molecular weights of GST-EFR3A (106.7 kDa) and His$_{10}$-KRAS$^{G12V}$ (22 kDa), these proteins eluted at volumes close to those of the protein standards globulin (16.27 ml; 158 kDa) and myoglobin (18.64 ml, 17 kDa) (Supplementary Fig. 5c). Importantly, the SEC profile of a preincubated mixture of GST-EFR3A with a two-fold excess of His$_{10}$-KRAS$^{G12V}$ revealed two main peaks, one at the elution volume of 15.29 ml and the other at 18.38 ml (Fig. 4q).

Immunoblot analysis with KRAS- and EFR3A-specific antibodies revealed that the first peak (15.29 ml) contained both His$_{10}$-KRAS$^{G12V}$ and GST-EFR3A, whereas the second peak (18.38 ml) contained only His$_{10}$-KRAS$^{G12V}$ (Fig. 4r). As His$_{10}$-KRAS$^{G12V}$ alone was exclusively detected at peak fractions at 18.49 ml, but not at 15.29 ml (Fig. 4o, p), our SEC analysis supports a direct interaction between GST-EFR3A and His$_{10}$-KRAS$^{G12V}$ and the formation of a stable protein complex.

To determine the specificity of EFR3A for active KRAS, we performed a competition assay with the known RAS effector BRAF using the above purified recombinant His$_{10}$-KRAS$^{G12V}$ and GST-EFR3A, and commercially purified recombinant BRAF. GST-EFR3A, His$_{10}$-KRAS$^{G12V}$, and increasing concentrations of BRAF were incubated, His$_{10}$-KRAS$^{G12V}$ was then immunoprecipitated, followed by immunoblot to detect GST-EFR3A. As controls, we demonstrate by immunoblot constant levels of His$_{10}$-KRAS$^{G12V}$ and increasing levels of BRAF in the mixture. This analysis revealed a stepwise reduction in the amount of GST-EFR3A that co-immunoprecipitates with His$_{10}$-KRAS$^{G12V}$ (Fig. 5a). These data are consistent with EFR3A having an affinity for the conformation of KRAS that binds effectors.

To map the region of EFR3A necessary to interact with KRAS$^{G12V}$, we generated four progressively larger C-terminal truncation mutants based on recombinant GST-EFR3A lacking the membrane targeting region (FL, Fig. 5b). Each truncation mutant was incubated with recombinant His$_{10}$-KRAS$^{G12V}$, after which His$_{10}$-KRAS$^{G12V}$ was immunoprecipitated, followed by immunoblot with KRAS- and EFR3A-specific antibodies. This revealed that deletion mutants lacking the terminal 122 to 138 amino acids (Δ1 and Δ2) were expressed but failed to co-immunoprecipitate with His$_{10}$-KRAS$^{G12V}$, while deletion mutants lacking the 22 to 109 C-terminal amino acids (Δ3 and Δ4) were co-immunoprecipitated (Fig. 5c, e). To address whether the C-terminus was alone sufficient to bind KRAS$^{G12V}$, we repeated the experiment with a GST-tagged polypeptide encoding the terminal 138 amino acids of EFR3A (mutant C, Fig. 5b), finding that this fragment co-immunoprecipitated with His$_{10}$-KRAS$^{G12V}$ (Fig. 5d, e). These data suggest that the region between amino acids 700 to 712 promotes the binding of GST-EFR3A$^{FL}$ to His$_{10}$-KRAS$^{G12V}$ in vitro, supporting the contention of a specific association of EFR3A to active KRAS.

**Endogenous EFR3A co-immunoprecipitates with endogenous active KRAS**. To address whether endogenous EFR3A binds active KRAS, we ectopically expressed FLAG epitope-tagged KRAS$^{G12D}$ (FLAG-KRAS$^{G12D}$) in human 293T cells. FLAG-KRAS$^{G12D}$ was then immunoprecipitated with a FLAG-specific antibody followed by immunoblot with an EFR3A-specific antibody. This revealed the co-immunoprecipitation of endogenous EFR3A with oncogenic KRAS (Fig. 5f, g). To confirm that this interaction is dependent upon the activation status of endogenous

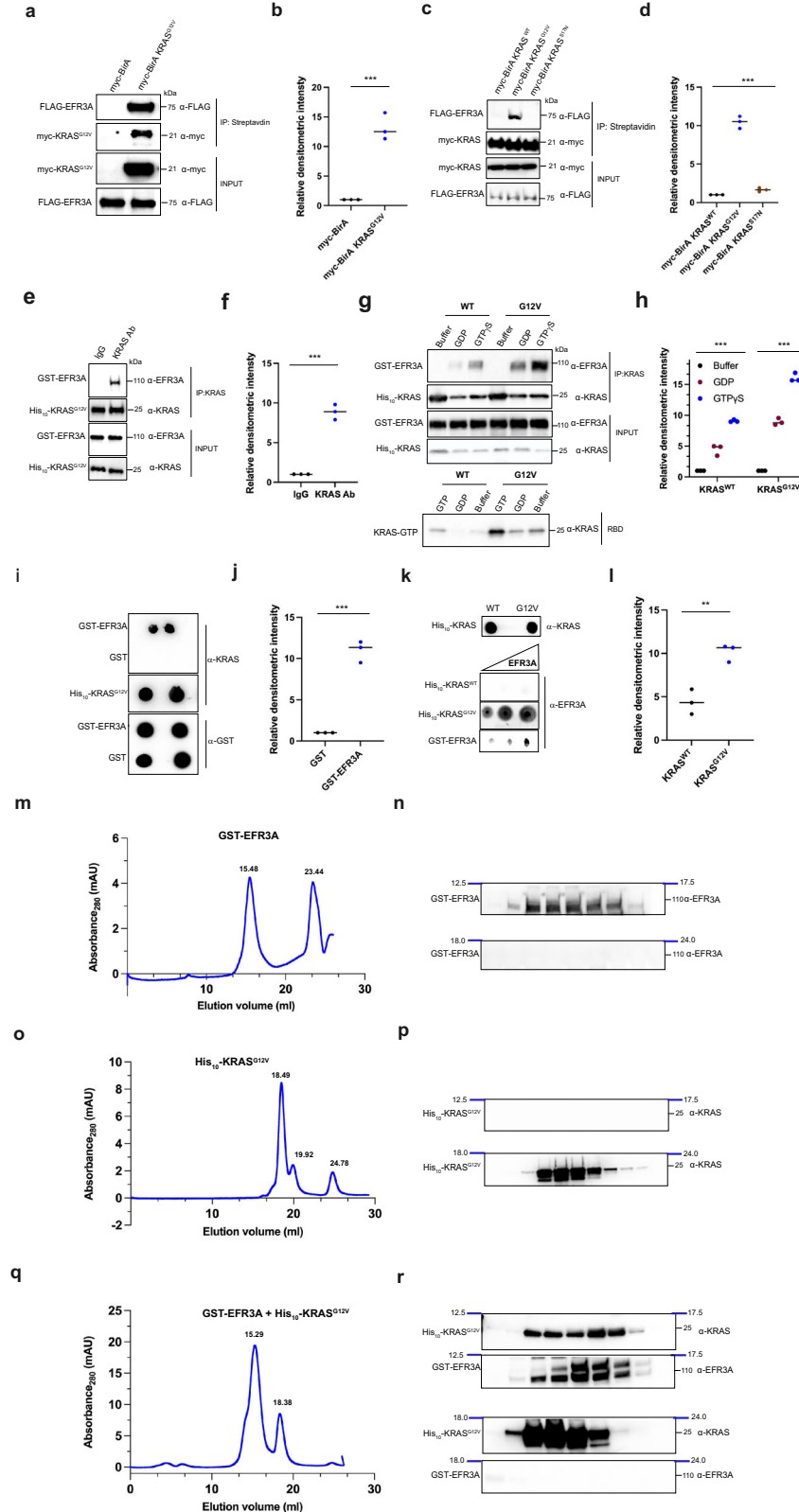

KRAS, we used the aforementioned RBD polypeptide to affinity capture endogenous GTP-bound RAS from lysates isolated from the KRAS mutation-positive human pancreatic adeno-carcinoma cell line AsPC-1. Immunoblot with an EFR3A-specific antibody detected endogenous EFR3A (Fig. 5h). Thus, endogenous EFR3A co-immunoprecipitates with endogenous GTP-bound KRAS in cells. The specificity of this interaction

was validated by demonstrating a temporal enrichment of EFR3A in the affinity-captured RAS-GTP pool after EGF stimulation of the KRAS mutation-negative BxPC3 cell line (Supplementary Fig. 5f). Collectively, these results support the contention that EFR3A directly binds KRAS in the active GTP-bound state in a fashion resembling how activated RAS engages with its effectors.

**Fig. 4 EFR3A associates with oncogenic RAS. a–d** Analysis of (**a, c**) the levels and (**b, d**) quantification of biotin labeling of FLAG-EFR3A expressed in 293T cells by ectopic myc-BirA*-KRAS wild-type (WT), G12V, or S17N or by myc-BirA alone as a control, as assessed by streptavidin affinity capture (IP) of biotinylated proteins and immunoblotted with FLAG- and myc-specific antibodies. INPUT levels are shown as loading controls. **e–h** Analysis of (**e, g**) the levels and (**f, h**) quantification of recombinant GST-EFR3A detected by immunoblot with an EFR3A-specific antibody that co-immunoprecipitates (co-IP) WT or G12V recombinant $His_{10}$-KRAS using a KRAS-specific antibody, and (**g, h**) preincubated with buffer, GDP, or GTPγS. INPUT levels are shown as loading controls. IgG serves as a negative control. **i, j** Analysis of (**i**) the levels and (**j**) quantification of recombinant $His_{10}$-$KRAS^{G12V}$ (preloaded with GTPγS) detected with a KRAS-specific antibody, that associates with recombinant GST-EFR3A versus control GST by far western. INPUT levels are shown as loading controls. **k, l** Analysis of (**k**) the levels and (**l**) quantification of GST-EFR3A detected with an EFR3A-specific antibody, that associate with recombinant wild-type (WT) versus $His_{10}$-$KRAS^{G12V}$ by far western. INPUT levels are shown as loading controls. **m–r** Analysis by (**m, o, q**) size-exclusion chromatography (SEC) profile and (**n, p, r**) immunoblot of the indicated fractions of recombinant (**m, n**) GST-EFR3A (peak 15.48: GST-EFR3A, **o, p** recombinant $His_{10}$-$KRAS^{G12V}$ preloaded with GTPγS (peak 18.49: $His_{10}$-$KRAS^{G12V}$), and (**q, r**) a complex of recombinant GST-EFR3A and $His_{10}$-$KRAS^{G12V}$ preloaded with GTPγS (peak 15.29: complex). **a–k** Representative of 3 biological independent samples tested in 3 independent experiments. **m–r** Representative of 3 biological replicates. Replicate experiments and full-length gels are provided in Supplementary Fig. 12. Significance values calculated by one-sided student's $t$ test for (**b, d, f, h, j, l**): **$p < 0.01$ and ***$p < 0.001$. Specific $p$ values for (**b**) is 0.00089, **d** are 0.00074 (G12V) and 0.00014 (S17N), **f** is 0.00058, (**h**) are 0.00048 ($WT^{GDP}$), 0.00015 ($WT^{GTP}$), 0.00031 ($G12V^{GDP}$), and 0.00028 ($G12V^{GTP}$), **j** is 0.00067, and **l** is 0.0069.

## Loss of EFR3A reduces KRAS plasma membrane localization and nanoclustering

Two observations suggest a direct effect of EFR3A on KRAS itself. First, as noted above, EFR3A bound to activated KRAS. Second, the loss of EFR3A reduced not only PI3K/AKT signaling—not unexpected as the protein recruits PI4K to the plasma membrane[28,29]—but also MAPK signaling. Such a finding suggests that EFR3A functions where these two pathways converge, namely at the KRAS oncoprotein. As KRAS must reside at the plasma membrane to signal[50], the very membrane whereby EFR3A resides[29], we explored the effect of EFR3A loss on the localization of KRAS at this membrane. Specifically, we transiently transfected vector or *sgEFR3A/B* HEK-HT cells with a vector encoding both GFP-$KRAS^{G12V}$ to visualize the subcellular localization of KRAS, and mCherry-CAAX, which marks endomembranes[24]. Confocal microscopy imaging revealed the expected distribution of GFP-$KRAS^{G12V}$ preferentially at the plasma membrane rather than endomembranes marked by mCherry-CAAX, while loss of EFR3A/B reversed this localization, with GFP-$KRAS^{G12V}$ preferentially detected on endomembrane (Fig. 6a). Quantification revealed a significant 3.4-fold shift of GFP-$KRAS^{G12V}$ from the plasma membrane to internal membranes (Fig. 6b). To independently validate these results with a completely different detection platform, we turned to biochemical fractionation. Specifically, $KRAS^{G12V}$-transformed HEK-HT cells in which *EFR3A* was knocked out, as confirmed by immunoblot (Fig. 2a), were subjected to membrane fractionation to isolate the plasma membrane fraction, as confirmed by the presence of the $Na^+/K^+$ ATPase by immunoblot[51], from the cytosolic fraction, as confirmed by the presence of GAPDH[52] (Fig. 6c). Immunoblot analysis revealed that the normal enrichment of KRAS in the membrane fraction was substantially reduced upon loss of EFR3A (Fig. 6c). To validate these results at the endogenous level, we repeated the experiment in the *KRAS*-mutant pancreatic adenocarcinoma cell line CFPAC-1, except we immunoblotted for endogenous KRAS protein. Again, in cells lacking EFR3A (Fig. 3a), KRAS was mis-localized from the membrane to cytosolic fraction (Fig. 6c). We ruled out that the loss of KRAS at the plasma membrane was due to an overall reduction in KRAS protein, as immunoblot quantification revealed a negligible change in the level of ectopic or endogenous KRAS protein upon loss of EFR3A (Fig. 6d).

We independently validated these results at the resolution of electron microscopy (EM). Specifically, vector, *sgEFR3A*, *sgEFR3B*, or *sgEFR3A/B* HEK-HT cells were transiently transfected with GFP-$KRAS^{G12V}$. Intact basal plasma membrane sheets were prepared from these cells and labeled with a GFP-specific antibody directly coupled to gold nanoparticles and imaged by a transmission electron microscopy (TEM). Loss of either or both

EFR3A or EFR3B significantly decreased the amount of immunogold-labeled $KRAS^{G12V}$ quantified by measuring gold particle density (Fig. 6e and Supplementary Fig. 6a), indicative of KRAS mis-localization from the inner plasma membrane. Specifically, there was a ~57% reduction in $KRAS^{G12V}$ at the plasma membrane in EFR3A- or EFR3B- knockout cells, and a 65% reduction in EFR3A/B double-knockout cells (Fig. 6f and Supplementary Fig. 6a). In addition to localizing to the plasma membrane, KRAS molecules form nanoclusters to promote signaling[19]. We therefore extended the analysis of the immunogold-labeled plasma membrane sheets to include spatial mapping of the nanogold point patterns. 20 to 30 images per sample were analyzed using the univariate K-function expressed as $L(r)$-$r$. Values of $L(r)$-$r$ greater than 1 (=99% CI for a random pattern) indicate significant clustering, and the peak value of the function termed $L_{max}$ serves as a useful summary statistic that quantifies the extent of clustering[23]. Loss of one or more EFR3 isoforms significantly reduced $KRAS^{G12V}$ nanoclustering, as evidenced by the reduction in $L_{max}$ values (Fig. 6g). Thus, by three independent assays, at both the ectopic and endogenous level, and by knocking out either or both EFR3 isoforms, we demonstrate that EFR3A promotes KRAS plasma membrane localization, nanoclustering, and engagement with effectors.

The above findings predict that other components of the EFR3A complex should similarly co-cluster with KRAS. To this end, 293T cells were transiently co-transfected with GFP-$KRAS^{G12V}$ and mCherry-EFR3A, mCherry-EFR3B, mCherry-TTC7A, or mCherry-FAM126A. 293T cells co-transfected with GFP-$KRAS^{G12V}$ and RFP-$KRAS^{G12V}$, or GFP-$KRAS^{G12V}$ and RFP-tH (the minimal C-terminal anchor of HRAS) served as positive and negative controls for proteins that extensively co-cluster or spatially segregate, respectively. Intact basal plasma membrane sheets prepared from these cells and attached to gold EM grids were fixed and co-labeled with GFP-specific and RFP-specific antibodies directly conjugated to 6 nm and 2 nm gold nanoparticles, respectively. The plasma membrane sheets were imaged in by TEM and co-localization between the 6 nm and the 2 nm gold particle distributions was quantified using bivariate K-functions expressed as $L_{biv}(r)$-$i$. As a summary parameter to quantify the extent of co-clustering, we use a defined integral of the $L_{biv}(r)$-$r$ function termed $L_{biv}$-Integrated (LBI). LBI values <100 (the 95% C.I.) indicate no significant co-clustering, whereas the higher the LBI value the more extensive the co-localization of the 2 nm and 6 nm gold distributions[11,23]. This analysis revealed that EFR3A, EFR3B, TTC7A, and FAM126A all co-cluster with $KRAS^{G12V}$ to an extent that is similar to GFP-$KRAS^{G12V}$ co-clustering with RFP-$KRAS^{G12V}$ (Supplementary Fig. 7). Conversely, the low LBI value for RFP-$KRAS^{G12V}$ and GFP-tH

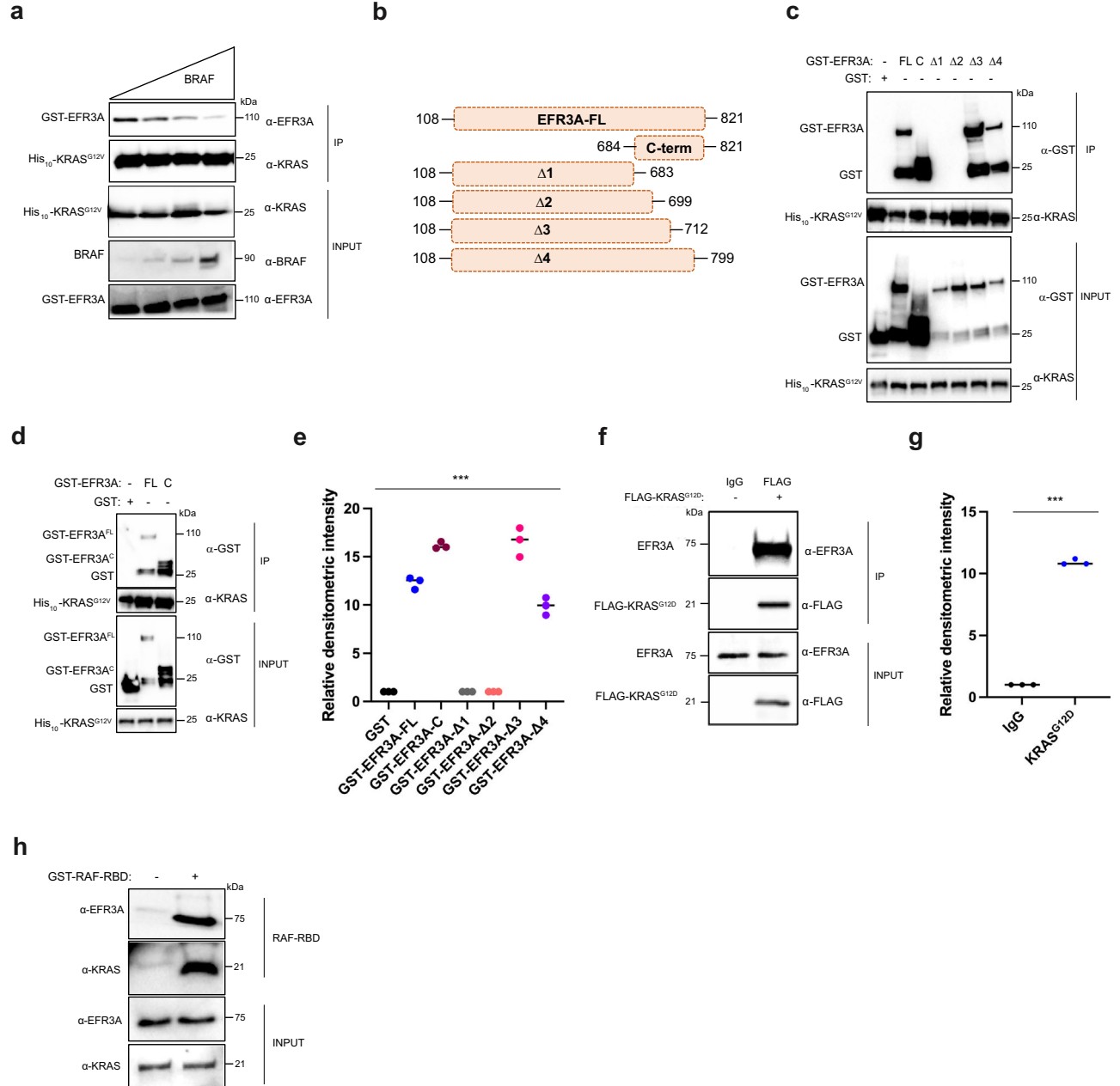

**Fig. 5 EFR3A associates with recombinant and endogenous oncogenic RAS. a** Analysis of levels of recombinant GST-EFR3A, as detected by immunoblot with an anti-EFR3A antibody, that co-immunoprecipitates (CO-IP) with recombinant His$_{10}$-KRAS$^{G12V}$ (preloaded with GTPγS) using an anti-KRAS antibody, in the presence of increasing levels of the RBD of BRAF, detected with and a BRAF-specific antibody. INPUT levels are shown as loading controls. **b–e** A (**b**) schematic of GST-EFR3A constructs used (FL: EFR3A base construct lacking 1-107 residues, C: C-terminal fragment, Δ: C-terminal truncations of FL) and (**c**, **d**) the levels and (**e**) quantification thereof of the indicated recombinant GST-EFR3A proteins, as detected by immunoblot with an EFR3A-specific antibody, co-immunoprecipitating (CO-IP) with recombinant His10-KRAS$^{G12V}$ (preloaded with GTPγS) using a KRAS-specific antibody. INPUT and IP serve as loading controls. GST serves as a negative CO-IP control. **f**, **g** Analysis of the (**f**) levels and (**g**) quantification thereof of the endogenous EFR3A, as detected by immunoblot with an EFR3A-specific antibody, co-immunoprecipitating (CO-IP) with FLAG-KRAS$^{G12D}$ using a FLAG-specific antibody when transiently expressed in 293T cells. **h** Levels of endogenous EFR3A bound to endogenous active KRAS, as assessed by RBD pull-down assay in AsPC-1 cells, as assessed by incubating AsPC-1 cell lysate with recombinant GST-RAF-RBD protein, followed by immunoblot with a KRAS-specific antibody to detect active KRAS and an EFR3A-specific antibody to detect endogenous EFR3A. INPUT and IP levels are shown as loading controls. **a**, **c**–**h** Representative of 3 biological replicates. Replicate experiments and full-length gels are provided in Supplementary Fig. 12. Significance values calculated by one-sided student's *t* test for (**e**, **g**): \*\*\**p* < 0.001. Specific *p* values for (**e**) are 0.00052 (FL), 0.00035 (C), 0.00011 (Δ1), 0.00014 (Δ2), 0.00056 (Δ3), and 0.00034 (Δ4) and (**g**) is 0.00087.

illustrates that proteins ectopically expressed on the plasma membrane do not always co-localize on the nanoscale. Collectively these data support the recruitment of EFR3A to activated KRAS increasing the retention and nanoclustering of KRAS at the plasma membrane.

**Targeting KRAS to the plasma membrane by an alternative anchor rescues the loss of EFR3A.** The above observations are consistent with a model whereby activated KRAS binds EFR3A, which retains KRAS at the plasma membrane to signal. A direct test of this hypothesis is that maintaining KRAS at the plasma

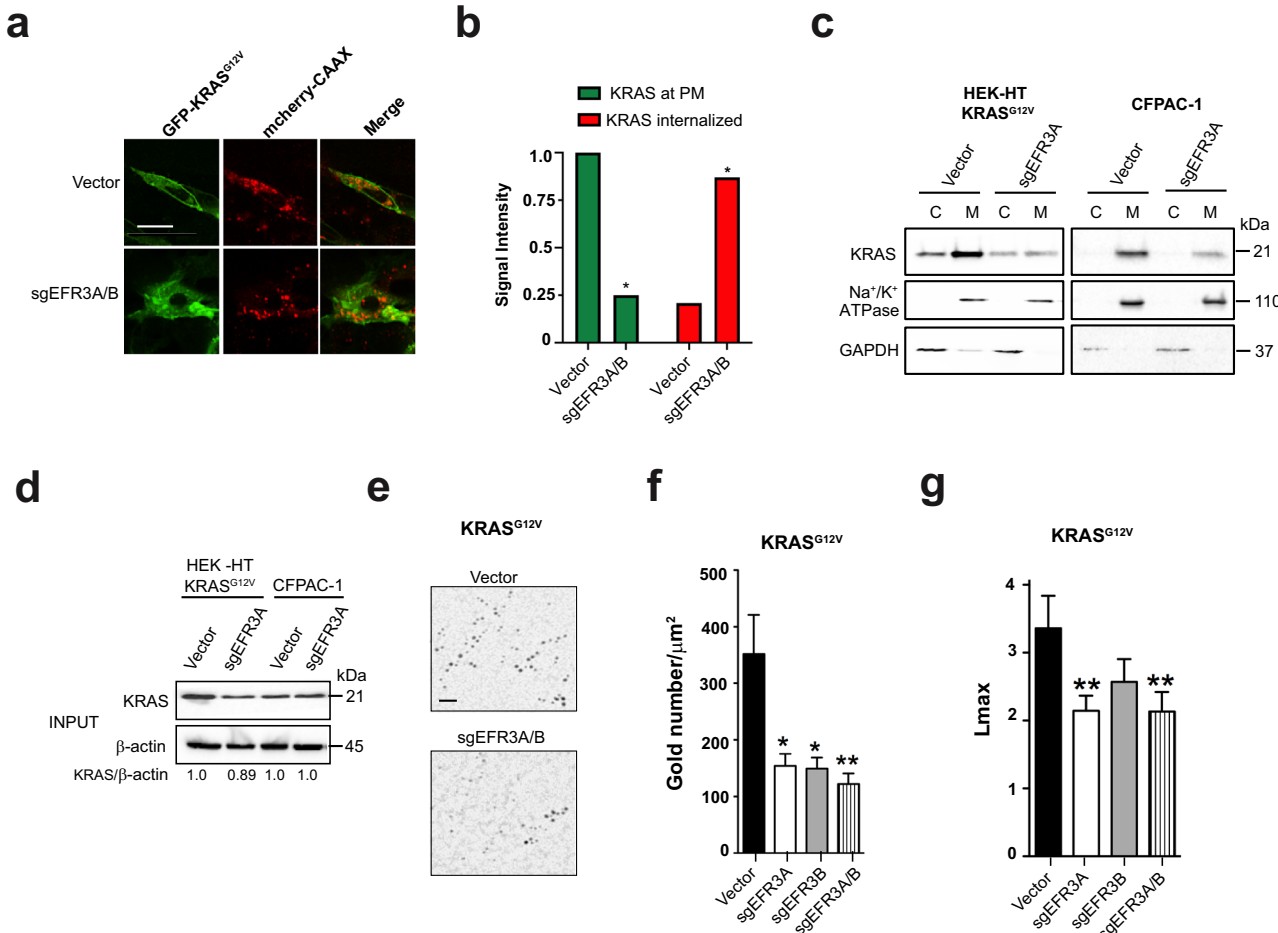

**Fig. 6 EFR3A promotes the localization and nanoclustering of KRAS at the plasma membrane. a** Representative confocal images showing the distribution of ectopically expressed GFP-KRAS$^{G12V}$, mCherry-CAAX, or both protein (merge) in HEK-HT cells stably infected with a lentiviral vector encoding Cas9 and no sgRNA (vector) or *sgEFR3A/B*. Scale bar: 20 μm. **b** Mean ± SD signal intensity of GFP-KRAS$^{G12V}$ at the plasma membrane (KRAS at PM) versus internalized (KRAS internalized) in the same two cell lines. **c** Levels of KRAS and Na$^+$/K$^+$ ATPase (PM marker), and GAPDH (cytosolic marker), as assessed by immunoblot analysis, in the cytosolic (C) versus membrane (M) fraction of KRAS$^{G12V}$-transformed human HEK-HT cells or the human pancreatic adenocarcinoma cell line CFPAC-1 stably infected with a lentivirus encoding Cas9 and no sgRNA (vector) or *sgEFR3A*. **d** Levels of KRAS versus actin as a loading control in the same cells from the endomembrane fraction. **e–g** A (**e**) representative EM micrograph showing gold conjugated, GFP-specific antibody (black dots) bound to GFP-KRAS$^{G12V}$, (**f**) mean ± SEM of gold particles per 1 μm$^2$ of plasma membrane sheets representative of the amount of KRAS on the plasma membrane, and (**g**) the extent of nanoclustering of KRAS on the plasma membrane as measured by *Lmax* from human HEK-HT cells transiently expressing GFP-KRAS$^{G12V}$ and stably infected with a lentivirus(es) encoding Cas9 with no sgRNA (vector) or the indicated sgRNAs. Scale bar: 100 nm. **a–d** Representative of 3 biological replicates. **e, f** EM analysis is based on 30 images. Replicate experiments and full-length gels are provided in Supplementary Fig. 13. Significance values were calculated by one-sided student's *t* test for (**b**) and two-sided bootstrap test for (**f, g**): \**p* < 0.05 and \*\**p* < 0.01. Specific *p* values for (**b**) are 0.011 (*sgEFR3A/B* KRAS at PM) and 0.024 (*sgEFR3A/B* KRAS internalized), **f** are 0.0138698 (*sgEFR3A*), 0.0117821 (*sgEFR3B*), and 0.005275 (*sgEFR3A/B*), **g** are 0.001 (*sgEFR3A*), 0.424 (*sgEFR3B*), and 0.001 (*sgEFR3A/B*).

membrane should rescue the defect in oncogenic KRAS signaling and transformation due to a loss of EFR3A. Localization of KRAS to the plasma membrane is mediated by a combination of a polybasic domain and prenylation of the C-terminal CAAX sequence[53]. As loss of EFR3A reduces the residency of KRAS at the plasma membrane, we queried whether targeting KRAS to the plasma membrane by other means overcomes this effect. To this end, we note that artificial N-terminal myristoylation can tether KRAS to the plasma membrane in the absence of prenylation[54]. We thus fused an N-terminal myristoylation sequence (myr)[55] to a SAAX-mutant (which blocks prenylation)[54] of oncogenic (G12V) myc epitope-tagged KRAS, creating myr-myc-KRAS$^{G12V}$. We note here that this construct still retains the polybasic domain of KRAS(4B) but lacks prenylation sites, which is known to be important for binding PS[23] that, as discussed below, is critical for retention of KRAS at the plasma membrane[11,23]. *sgEFR3A/B*

HEK-HT cells were stably infected with a lentivirus expressing myr-myc-KRAS$^{G12V}$. As controls, HEK-HT cells with and without *sgEFR3A/B* were infected with a lentivirus expressing myc-KRAS$^{G12V}$. The three cell lines were then subjected to the same biochemical fractionation and immunoblot analysis as above. This revealed that myr-myc-KRAS$^{G12V}$ was enriched in the membrane fraction in the absence of EFR3A/B beyond that of control myc-KRAS$^{G12V}$ (Fig. 7a), and thus could be used for epistatic analysis. To next address whether KRAS retention at the plasma membrane in this manner could rescue the loss of EFR3A/B, we analyzed these three cell lines for oncogenic KRAS signaling. Immunoblot analysis revealed that the decrease in P-ERK levels upon loss of EFR3A/B were restored by expressing myr-myc-KRAS$^{G12V}$, but not in the control myc-KRAS$^{G12V}$, back to the level of positive-control cells (Fig. 7b). We ascribe this to recruiting RAS effectors to the plasma membrane, as C-RAF was enriched in the

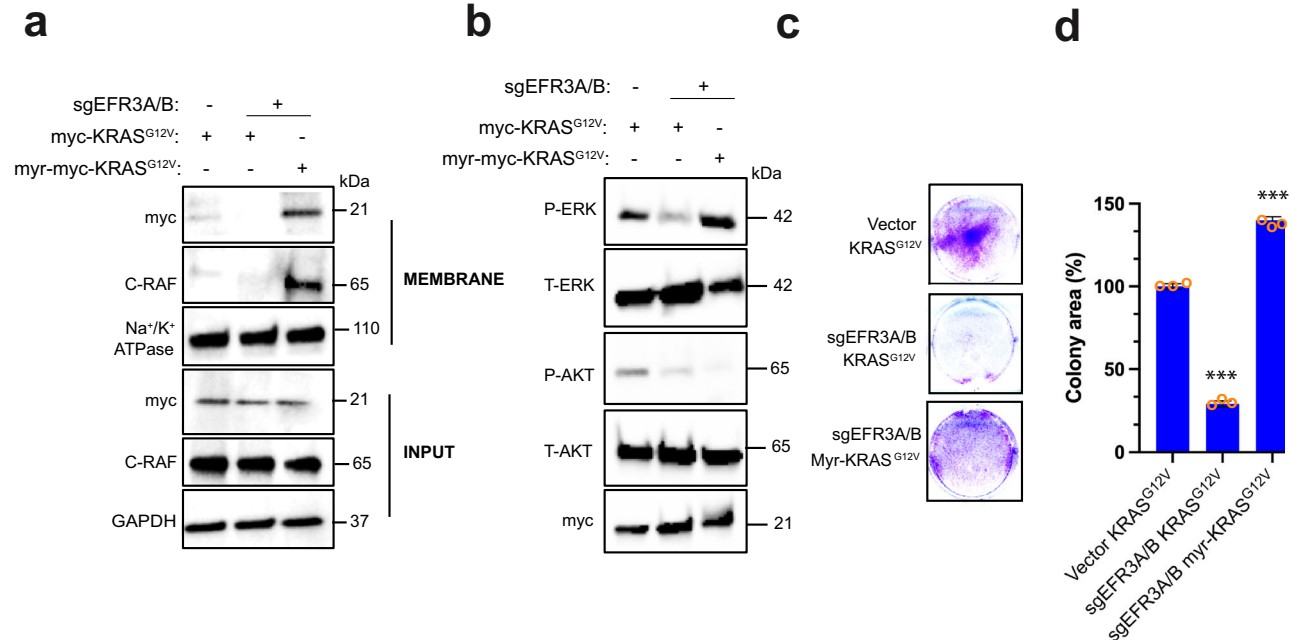

**Fig. 7 KRAS membrane anchoring rescues signaling and transformation defects in EFR3A-null cells. a–d** Immunoblot analysis of (**a**) membrane fraction for myc-KRAS[G12V], myr-myc-KRAS[G12V], C-RAF, Na+/K+ ATPase, and GAPDH, and (**b**) phosphorylated (P) and total (T) ERK and AKT, and (**c**) an example of colony formation, as assessed by crystal violet staining, which is (**d**) plotted as mean ± SD of colony area, for human HEK-HT cells stably infected with a lentivirus encoding Cas9 encoding no sgRNA (vector) or *sgEFR3A/B* in the absence and presence of ectopic myc-KRAS[G12V] or when targeted to the PM by an N-terminal myristoylation sequence (myr-myc-KRAS[G12V]). INPUT levels of indicated proteins serve as loading controls. **a, b** Representative of 3 biological replicates. **c, d** Representative of 3 biological replicates tested in triplicate. Replicate experiments and full-length gels are provided in Supplementary Fig. 13. Significance values were calculated by one-sided student's *t* test for (**d**): ***p < 0.001. Specific *p* values for (**d**) are 0.00027 (KRAS[G12V]*sgEFR3A/B*) and 0.00022 (myr-KRAS[G12V]*sgEFR3A/B*).

membrane fraction in cells lacking EFR3A/B upon specifically expressing myr-myc-KRAS[G12V] (Fig. 7a). Interestingly however, we note that myr-myc-KRAS[G12V] did not restore P-AKT levels, perhaps reflecting an inability of this protein to completely replicate the normal membrane localization of KRAS, the likelihood of a different lipid composition around myristoylated KRAS clusters, or the additional importance of recruiting EFR3A to generate PI(4)P required for PI3K activity. Finally, to address whether myr-myc-KRAS[G12V] could rescue the loss of transformation, these three cell lines were assayed for 2D-transformed growth, which revealed that the reduction of colony formation due the loss of EFR3A/B was rescued to the level of the control KRAS[G12V]-transformed cells upon expressing myr-myc-KRAS[G12V] but not myc-KRAS[G12V] (Fig. 7c, d). To evaluate the effect on 3D-transformed growth, these three cell lines were assayed for growth in soft agar, which similarly revealed restoration of anchorage-independent colony formation by myr-myc-KRAS[G12V] (Supplementary Fig. 8). Thus, epistatic analysis supports EFR3A promoting oncogenic KRAS signaling and transformation by, in large part, retaining the oncoprotein at the plasma membrane, likely through specific lipid species.

**Loss of EFR3A alters the phosphatidylserine content, KRAS localization, and nanoclustering at the plasma membrane.** KRAS nanoclustering and plasma membrane localization are abrogated by reducing the PS content at the plasma membrane[23,56]. PS is required for both the localization of KRAS to this membrane and nanoclustering, which is mediated by the exquisite binding specificity of the KRAS C-terminal membrane anchor for PS. PS is transported to the plasma membrane by the ORP5 and ORP8 lipid transporters that exchange PS for PI(4)P generated by PI4KA at the plasma membrane[24,26,57]. Since EFR3A functions to recruit PI4KA[28] to the plasma membrane[29] and knockout of EFR3A

reduced KRAS plasma membrane localization and nanoclustering, we addressed whether these effects were mediated by PI(4)P and PS levels on the plasma membrane. Vector, *sgEFR3A*, *sgEFR3B*, and *sgEFR3A/B* HEK-HT cells were transiently transfected with GFP-P4M-SidM. GFP-P4M-SidM is a PI(4)P biosensor comprised of GFP for detection purposes fused in-frame with the PI(4)P binding domain of the protein SidM[58] that specifically recognizes the plasma membrane pool of PI(4)P. As above, derived plasma membrane sheets were incubated with a GFP-specific antibody directly coupled to gold nanoparticles and imaged by TEM, which revealed approximately a 64% reduction of immunogold-labeled GFP-P4M-SidM upon the loss of one or both ERF3 isoforms (Fig. 8a, b and Supplementary Fig. 6b). Since PI(4)P is essential for PS localization to the plasma membrane[59,60], we addressed whether EFR3A ablation also reduced plasma membrane PS levels. The same HEK-HT cells were transiently transfected with the PS biosensor GFP-LactC2 in which GFP is fused in-frame the C2 domain encoding the PS-binding activity of the protein lactadherin[56]. Again, immuno-EM analysis revealed a decrease in immunogold-labeled GFP-LactC2 in the plasma membrane of cells lacking one or both EFR3 isoforms (Fig. 8c, d and Supplementary Fig. 6c). Importantly, addition of exogenous PS to cells prior to plasma membrane sheet preparation rescued the plasma membrane localization (Fig. 8e) and nanoclustering (Fig. 8f) of GFP-KRAS[G12V] expressed in *sgEFR3A/B* HEK-HT cells. We further confirm that this increase in KRAS[G12V] plasma membrane localization and nanoclustering was indeed a result of increased plasma membrane PS levels, as addition of PS to *sgEFR3A/B* HEK-HT cells transiently expressing GFP-LactC2 increased the amount of plasma membrane-localized GFP-LactC2 visualized by immuno-EM (Fig. 8g). Thus, we suggest that the loss of EFR3A reduces plasma membrane PI(4)P and PS levels leading to reduced plasma membrane localization and nanoclustering of KRAS.

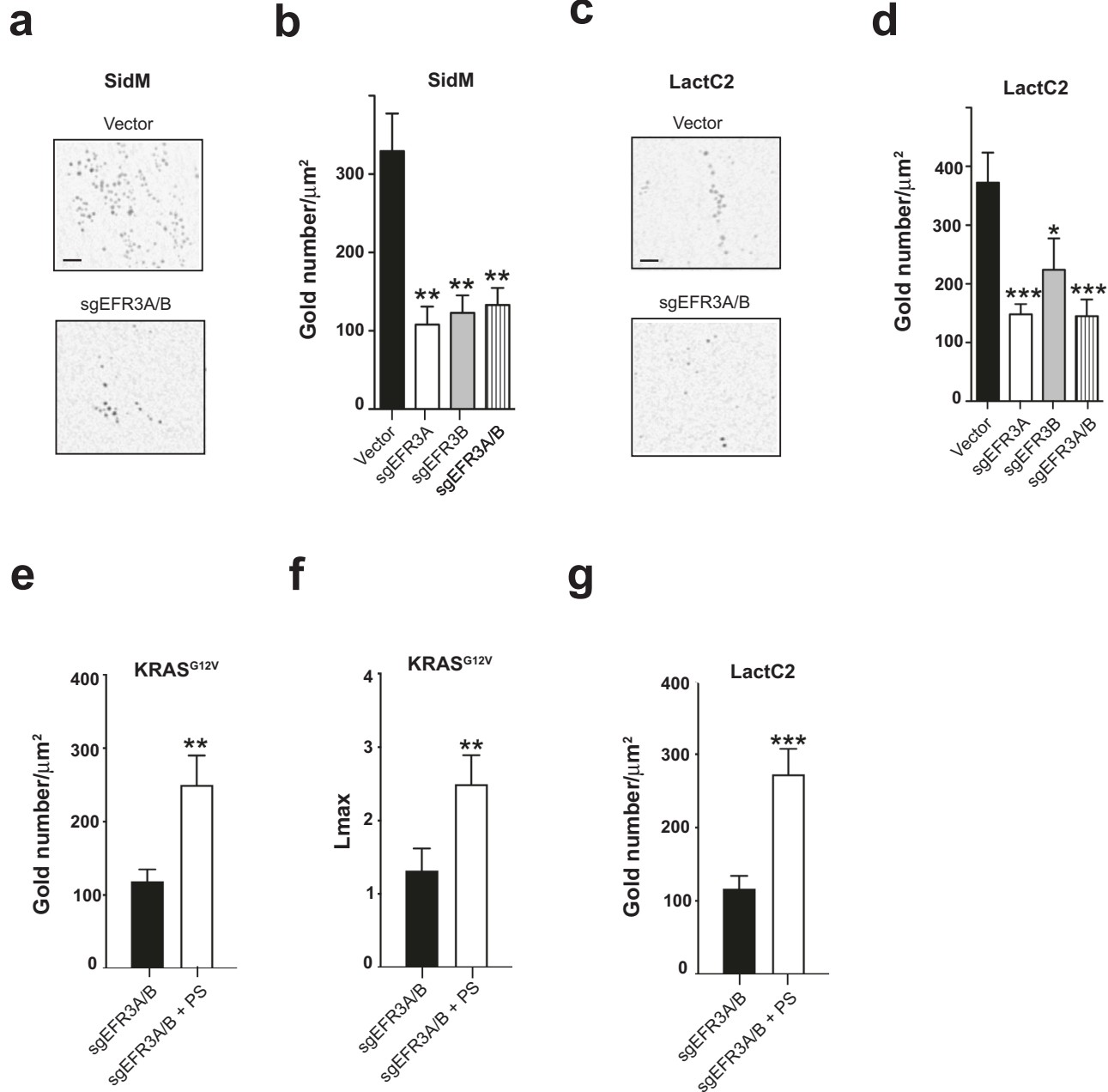

**Fig. 8 EFR3A regulates plasma membrane lipid content to promote RAS oncogenesis. a, c** Representative EM micrographs of plasma membrane (PM) sheets from human HEK-HT cells stably infected with a lentivirus(es) encoding Cas9 with no sgRNA (vector) or the indicated sgRNAs and transiently transfected with (**a**) GFP-P4M-SidM (a marker of PM PI(4)P) or (**c**) GFP-LactC2 (a marker of PM phosphatidylserine). Gold-conjugated GFP-antibodies seen as black dots are bound to the probe of interest. Scale bar: 100 nm. **b, d** The corresponding mean ± SEM of the number of gold particles per 1 μm² of plasma membrane sheets representative of amount of (**b**) PI(4)P and (**d**) PS on the plasma membrane. **e, f** Mean ± SEM of (**e**) gold particles per 1 μm² of plasma membrane sheets representative of the amount of KRAS on the plasma membrane and the (**f**) the extent of nanoclustering of KRAS on the plasma membrane as measured by *Lmax* from human HEK-HT cells transiently expressing GFP-KRAS^G12V and stably infected with a lentiviruses encoding Cas9 and *sgEFR3A/B* in the absence and presence of exogenous phosphatidylserine (PS). **g** Mean ± SEM of gold particles per 1 μm² of the plasma membrane sheets from human HEK-HT cells transiently transfected with GFP-LactC2 and stably infected with a lentiviruses encoding Cas9 and *sgEFR3A/B* in the absence and presence of exogenous phosphatidylserine (PS). **a–g** EM analysis is based on 30 images. Significance values were calculated by the two-sided bootstrap test for (**b, d–g**): \*$p < 0.05$, \*\*$p < 0.01$, and \*\*\*$p < 0.001$. Specific $p$ values for (**b**) are 0.00062274 (*sgEFR3A*), 0.00111426 (*sgEFR3B*), 0.00167654 (*sgEFR3A/B*), **d** are 0.000436671 (*sgEFR3A*), 0.056486805 (*sgEFR3B*), 0.000536135 (*sgEFR3A/B*), **e** is 0.00680191 (*sgEFR3A/B* + PS), **f** is 0.001 (*sgEFR3A/B* + PS), and (**g**) is 0.00051866 (*sgEFR3A/B* + PS).

**EFR3A promotes KRAS oncogenesis through PI4KA.** The above findings argue that EFR3A promotes oncogenic KRAS signaling and transformation by recruiting PI4KA to alter the local lipid composition of the plasma membrane. EFR3A forms a complex with TTC7A or B and FAM126A in addition to PI4KA[28]. To therefore determine if PI4KA acts downstream of EFR3A, we tested whether the loss of PI4KA affects oncogenic KRAS signaling and transformation. To this end, *KRAS*-mutant

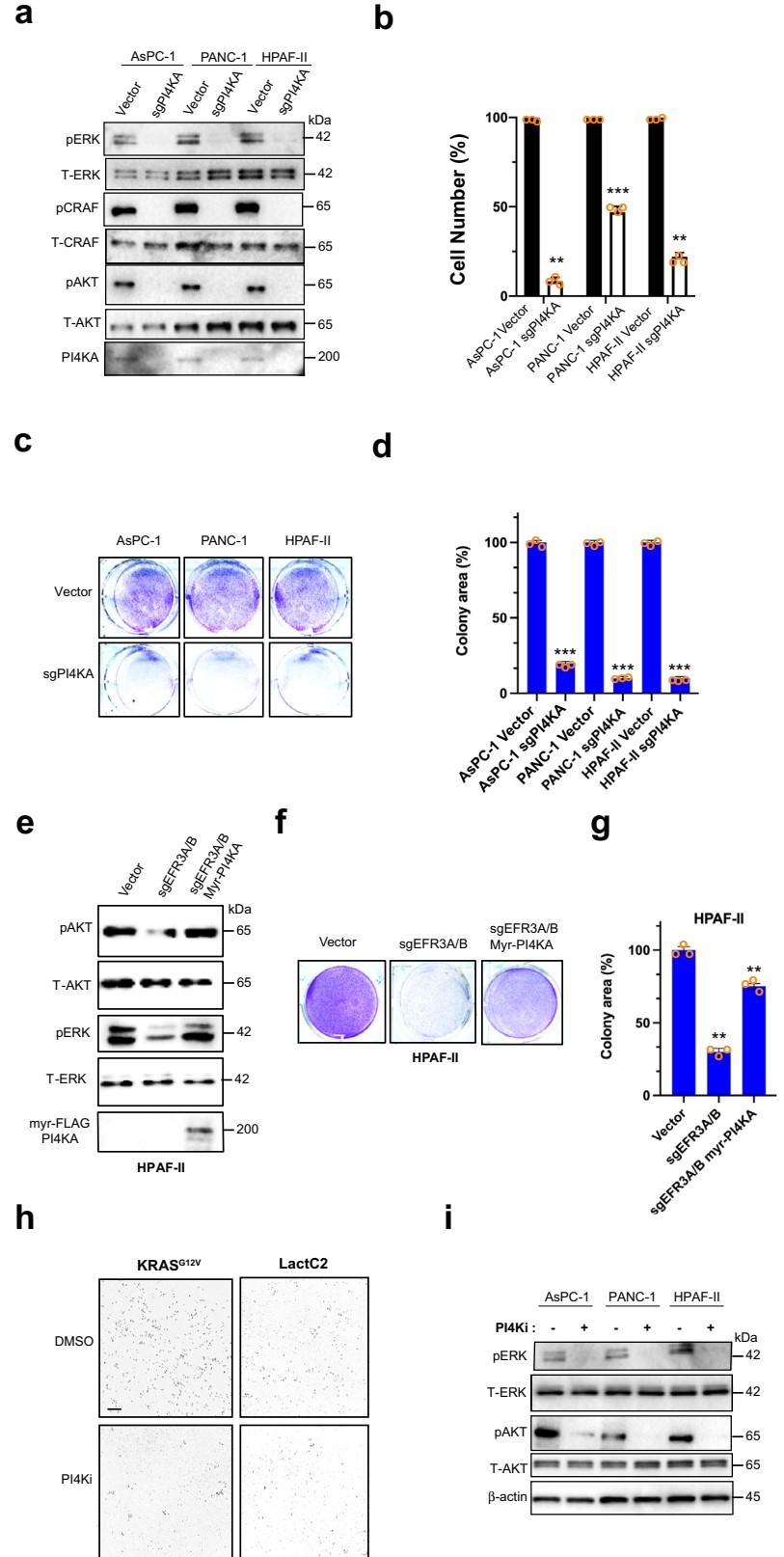

human pancreatic adenocarcinoma cell lines AsPC-1, PANC-1, and HPAF-II were stably transduced with a vector encoding Cas9 in the absence (vector control) and presence of an sgRNA targeting the *PI4KA* gene (*sgPI4KA*), which was confirmed by immunoblot to reduce endogenous PI4KA levels (Fig. 9a). The loss of PI4KA also reduced P-AKT, P-ERK, and P-CRAF levels

(Fig. 9a), cell number over time (Fig. 9b), 2D- (Fig. 9c, d), and 3D- (Supplementary Fig. 9a) transformed growth in all three cell lines. Given these results, we next performed epistatic analysis by testing whether tethering PI4KA to the plasma membrane could rescue the loss of endogenous oncogenic KRAS signaling and transformation in EFR3A/B-knockout cells. To this end, HPAF-II

**Fig. 9 PI4KA promotes plasma membrane association of oncogenic RAS. a–d** Analysis of the (**a**) level of phosphorylated (P) and/or total (T) ERK, AKT, and PI4KA, as assessed by immunoblot, **b** mean ± SD cell number over time in culture, as assessed by the Titer Glo assay, **c** an example of colony formation, as assessed by crystal violet staining, which is (**d**) plotted as the mean ± SD total colony area, in the *KRAS*-mutant human pancreatic adenocarcinoma cell lines AsPC-1, PANC-1, and HPAF-II stably infected with a lentivirus encoding Cas9 and no sgRNA (vector) or *sgPI4KA*. **e–g** Analysis of the (**e**) level of phosphorylated (P) and/or total (T) ERK, AKT, and PI4KA with an N-terminal myristoylation sequence and FLAG epitope-tag (myr-FLAG-PI4KA), as assessed by immunoblot, and (**f**) an example of colony formation, as assessed by crystal violet staining, which is (**g**) plotted as the mean ± SD total colony area, of HPAF-II cells stably infected with a lentivirus(es) encoding Cas9 and no sgRNA (vector) or *sgEFR3A/B* in the absence or presence of myr-FLAG-PI4KA. **h** A representative EM micrograph showing gold conjugated, anti-GFP antibody (black dots) reactivity in plasma membrane sheets from MDCK cells ectopically expressing GFP-KRAS$^{G12V}$ or GFP-LactC2 and treated with 30 μM C7 (PI4Ki) or vehicle (DMSO) for 48 h. Scale bar: 100 nm. **i** Levels of (P) and (T) ERK, AKT, and actin (as a loading control), as assessed by immunoblot analysis, in the human pancreatic adenocarcinoma cell lines AsPC-1, PANC-1, and HPAF-II after treatment with control DMSO or 20 μM of C7 (PI4Ki) for 4 h. **a, e, i** Representative of 3 biological replicates. **b–d, f, g** Representative of 3 biologic replicates tested in triplicate. **h** EM analysis is based on 30 images. Replicate experiments and full-length gels are provided in Supplementary Fig. 14. Significance values were calculated by one-sided student's t test (**b, d, g**): *$p < 0.05$, **$p < 0.01$, ***$p < 0.001$, and ****$p < 0.0001$. Specific *p* values for (**b**) are 0.0036 (AsPC-1*sgPI4KA*), 0.0003 (PANC-1*sgPI4KA*), and 0.0045 (HPAF-II*sgPI4KA*), **d** are 0.00063 (AsPC-1*sgPI4KA*), 0.00052 (PANC-1*sgPI4KA*), and 0.00021 (HPAF-II*sgPI4KA*), and (**g**) are 0.0017 (*sgEFR3A/B*) and 0.0065 (*sgEFR3A/B* myr-PI4KA).

---

cells in which EFR3A/B were knocked out (Fig. 3a) were engineered to stably express PI4KA fused to the previously described N-terminal myristoylation sequence (myr-PI4KA), a strategy that has been used to localize other PI-kinases to the plasma membrane[61]. Vector encoding no sgRNA or *sgEFR3A/B* served as positive and negative controls, respectively. Consistent with the above model, myr-PI4KA rescued P-ERK and P-AKT (Fig. 9e), 2D- (Fig. 9f, g), and 3D- (Supplementary Fig. 9b) transformed growth of HPAF-II cells lacking EFR3A/B to the levels observed in positive control cells. Thus, recruiting PI4KA to the plasma membrane in the absence of EFR3A stimulates oncogenic KRAS signaling and transformation. To pharmacologically affirm the role of PI4KA in KRAS signaling, we first confirmed that the selective PI4KA inhibitor C7[62] does indeed reduce PS and KRAS at the plasma membrane. As expected, plasma membrane sheets imaged by immunogold-EM analysis revealed a reduction of PS (as assessed with the biomarker Lact-C2) and KRAS$^{G12V}$ (GFP-KRAS$^{G12V}$) at the plasma membrane in MDCK cells (chosen for ease of imaging) treated with C7 at a dose of 30 μM for 48 h compared to cells treated with vehicle alone (Fig. 9h). Given this, we next assayed the effect of C7 on endogenous oncogenic KRAS signaling in *KRAS*-mutant human pancreatic adenocarcinoma cell lines. Specifically, the aforementioned AsPC-1, PANC-1, and HPAF-II cell lines were treated with C7 at a dose of 20 μM for 4 h or vehicle alone as a control, followed by immunoblot analysis, which revealed a robust reduction in P-AKT and P-ERK in cells treated with C7 (Fig. 9i). Collectively, these data are consistent with a feedback circuit whereby activated KRAS binds EFR3A, which in turn recruits PI4KA and thereby increases plasma membrane levels of PI(4)P and PS to promote the localization and nanoclustering of oncogenic KRAS at the plasma membrane, stimulating oncogenic signaling and transformation.

**Combined inhibition of oncogenic KRAS and PI4KA is synergistic.** The finding that EFR3A recruits PI4KA and binds oncogenic KRAS suggests the possibility of targeting this signaling circuit at the level of the kinase for the treatment of KRAS-mutant cancers. One challenge of pharmacologically targeting phosphatidylinositol kinases is the toxicity of inhibiting this class of enzymes[63], and PI4KA is no exception to this[63]. Indeed, monotreatment with a PI4KA inhibitor developed for the treatment of Hepatitis C infection exhibited acute toxicity in mice[63]. However, combining a PI4K inhibitor with radiation was found to decrease xenograft tumor growth[64,65]. We thus suggest that a PI4KA inhibitor may have a therapeutic window in situations whereby oncogenic KRAS signaling is already reduced, rendering cancer cells hypersensitive to any further loss of KRAS

signaling[8,63]. In this regard, we note that inhibitors targeting KRAS$^{G12C}$ are showing promise in the clinic, with sotorasib now approved for the treatment of lung cancer, but that drug combinations may be required for durable responses[8,9]. Given this, we treated three different KRAS$^{G12C}$-mutant human cancer cell lines (MiaPaCa-2, H2030, and H358) in triplicate with the vehicle DMSO as a control, the KRAS$^{G12C}$ inhibitor AMG510 (sotorasib)[4] at a fixed dose of 5 nM, which corresponds to an IC$_{20}$ to IC$_{30}$ depending on the cell line, and the PI4KA inhibitor C7[62] at a fixed dose of 10 nM, which corresponds to the approximate IC$_{30}$ to IC$_{55}$, depending on the cell line, or both compounds at these concentrations. 72 h later the number of viable cells was calculated by the Titer Glo assay and normalized to those of the vehicle-treated control cells, which was expressed as the percent of viable cells. This revealed that the combination of the KRAS$^{G12C}$ and PI4KA inhibitors was uniformly more inhibitory, reducing the number of viable cells by 80% to 95% (Fig. 10a). To assess the effect of protracted inhibitor treatment, we repeated the experiment, but at the lower IC$_{10}$ dose and for 12 consecutive days using colony formation as metric of 2D-transformed growth. This analysis revealed potent synergy between the two inhibitors, leading to an almost complete loss of colony formation (Fig. 10b, c). This is unlikely a general effect of disrupting PI4K activity, as reducing EFR3A/B expression (Supplementary Fig. 10a) was synergistic with AMG150 in two of these three lines (Supplementary Fig. 10b). Given these results, we expanded the doses to 0, 5, 10, 20, and 40 nM of AMG510 and 0, 10, 20, and 30 nM of C7 to derive a dose inhibition matrix[66]. This again revealed a synergistic effect across at a wide range of doses, but particularly at low doses that may be better tolerated clinically (Fig. 10d, f, h). To quantitate the degree of synergy we calculated the Bliss synergy score across all doses, which was 11.92 for MiaPaca-2, 10.13 for H2030, and 10.65 for H358 (Fig. 10e, g, i). Thus, low doses of a PI4KA inhibitor greatly augment the antineoplastic activity of a KRAS$^{G12C}$ inhibitor, suggesting a potential clinical application.

## Discussion
Here we identify EFR3A as a component of the RAS interactome mediating a positive-feedback circuit that promotes oncogenic KRAS signaling and tumorigenesis, unearthing a potential vulnerability to enhance the antineoplastic activity of KRAS$^{G12C}$-inhibitors for the treatment of pancreatic and possibly other KRAS-mutant cancers. Mechanistically, we propose that activated KRAS binds EFR3A, recruiting PI4KA and increasing the local concentration of PI(4)P and in turn PS, which maintains oncogenic KRAS localization and nanoclustering at the plasma membrane,

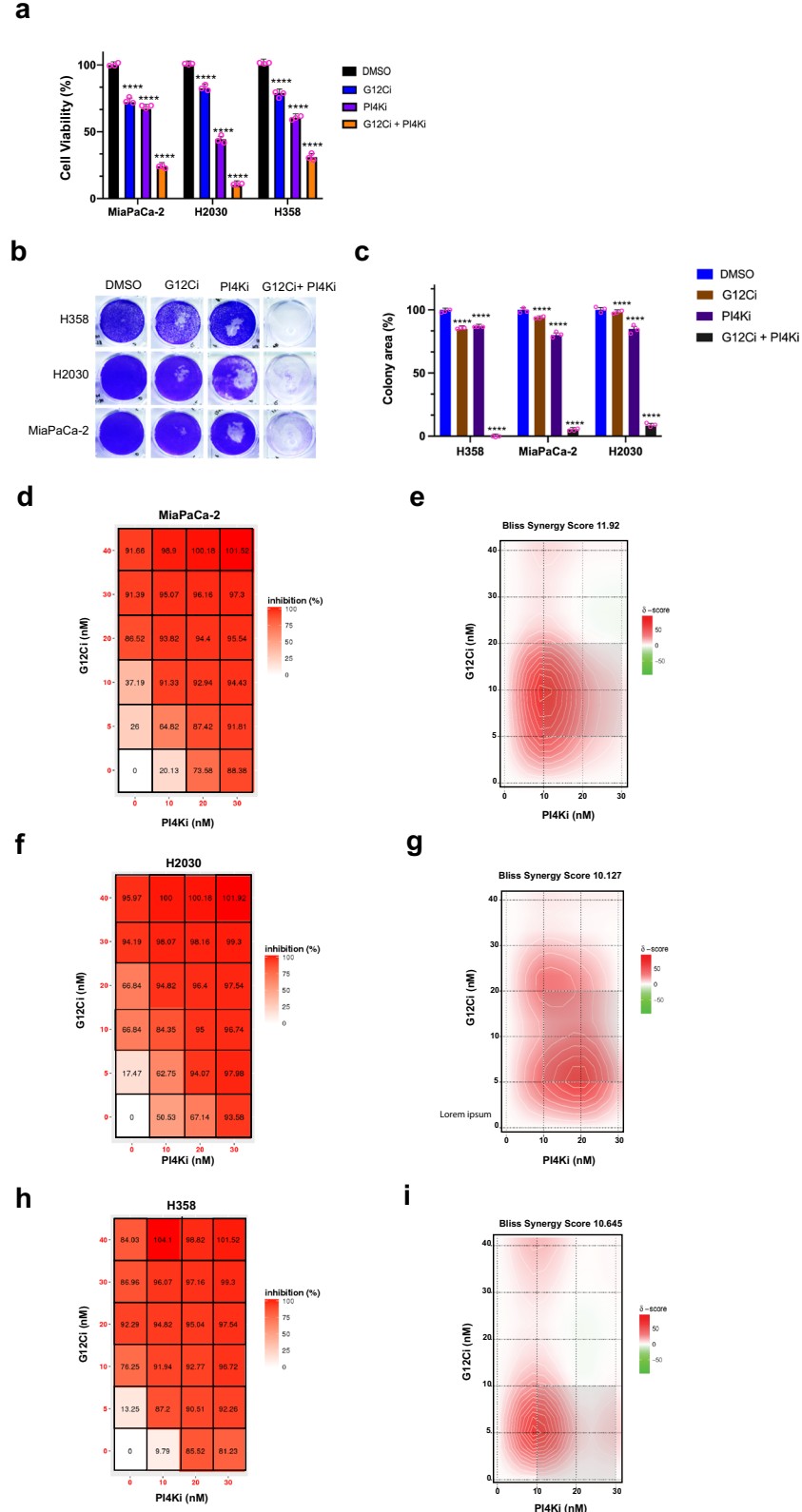

leading to sustained recruitment of KRAS effectors and signaling that promotes tumorigenesis (Supplementary Fig. 16).

The first step of this model, EFR3A preferentially binds KRAS in the active state, is supported by multiple experiments. Namely, we show that recombinant EFR3A binds to recombinant GTP-bound and/or oncogenic, but not GDP-bound and/or mutant-inactive KRAS, as assessed by co-immunoprecipitation and far western analysis. This association is disrupted by a RAS effector, and an EFR3A/KRAS$^{G12V}$ complex is captured by analytical size-exclusion chromatography. We also mapped the binding region on EFR3A to 12 amino acid stretch in the C-terminus. Of note, the C-terminus is also responsible for binding TTC7A, which stabilizes the EFR3/TTC7/FAM126/PI4KA complex on the plasma membrane[30–32]. The association of EFR3A with active

**Fig. 10 A PI4KA inhibitor synergized with a clinical RAS$^{G12C}$ inhibitor. a** Mean ± SD cell viability, as assessed by the Cell Titer Glo assay, of the *KRAS*$^{G12C}$-mutant human cancer cell lines MiaPaCa-2, H2030, and H358 treated with a fixed concentration of vehicle (DMSO), 5 nM of the RAS$^{G12C}$ inhibitor (G12Ci) AMG510, 10 nM of the PI4KA inhibitor (PI4Ki) C7, or both drugs at these concentrations. **b, c** An example of (**b**) colony formation, as detected by crystal violet staining, which is (**c**) plotted as the mean ± SD of colony area, of MiaPaCa-2, H2030, and H358 cells treated with the indicated drugs at a IC$_{10}$ dose for 12 days. **d, f, h** Dose response inhibition matrix and (**e, g, i**) BLISS synergy plot of (**b, c**) MiaPaCa-2, **d, e** H2030, and (**f, g**) H358 cells treated with the indicated drug concentrations for 72 h. **a–i** Representative of 3 biological replicates assayed in triplicate. Replicate experiments are provided in Supplementary Fig. 15. Significance values were calculated by two-sided Welch's *t*-test (**a, c**): ****$p < 0.0001$. Specific *p* values for (**a**) are 0.000021 (G12Ci), 0.000036 (PI4Ki), and 0.000047 (G12Ci + -PI4Ki) for MiaPaCa-2 cells, 0.000043 (G12Ci), 0.000034 (PI4Ki), and 0.000038 (G12Ci + PI4Ki) for H2030 cells, and 0.000013 (G12Ci), 0.000021 (PI4Ki), and 0.000019 (G12Ci + PI4Ki) for H358 cells, for **c** are 0.000021 (G12Ci), 0.000036 (PI4Ki), and 0.000032 (G12Ci + -PI4Ki) for MiaPaCa-2 cells, 0.000025 (G12Ci), 0.000016 (PI4Ki), and 0.000033 (G12Ci + PI4Ki) for H2030 cells, and 0.000038 (G12Ci), 0.000024 (PI4Ki), and 0.000017 (G12Ci + PI4Ki) for H358 cells.

KRAS was also captured in cells. Namely, we show a stepwise increase in biotin labeling of EFR3A by progressively more active BirA-KRAS in cells, and further, endogenous GTP-bound KRAS was associated endogenous EFR3A. Collectively, these data suggest that EFR3A is recruited to active GTP-bound KRAS in a manner reminiscent of effector proteins.

The second step of this model is that EFR3A recruits PI4KA, thereby increasing the local concentration of PI(4)P and PS. In this regard, we note that it is well established that EFR3A recruits PI4KA to the plasma membrane. Indeed, we see co-localization of all components of the EFR3A complex (including PI4KA) with KRAS$^{G12V}$ at the plasma membrane, and that a loss of EFR3A or PI4KA reduces PI(4)P levels at the plasma membrane, as assessed by immuno-EM. Further, we previously demonstrated that depleting PI(4)P on the plasma membrane results in the loss of the exchange of PI(4)P with PS in the endoplasmic reticulum, thereby reducing PS at the plasma membrane[24]. Again, we show that loss of EFR3A or PI4KA also reduces the level of PS at the plasma membrane.

The third step of this model is that the accumulation of PI(4)P and PS promotes active KRAS accumulation and nanoclustering at the plasma membrane[23,24]. In support, we previously demonstrated that the KRAS membrane anchor selectively interacts with symmetric unsaturated PS species (such as 16:1/18:1) to promote plasma membrane binding and asymmetric unsaturated PS (such as 16:0/18:1) to form nanoclusters[23]. Here we show that loss of EFR3A, EFR3B, and/or PI4KA reduces the amount of KRAS$^{G12V}$ at the plasma membrane, as assessed by immuno-fluorescence, biochemical fractionation, and/or immuno-EM. Similarly, we show that the loss of any of these proteins reduces KRAS$^{G12V}$ nanoclustering. Finally, we show that tethering PI4KA to the plasma membrane or providing PS to cells restores KRAS$^{G12V}$ localization and nanoclustering at the plasma membrane in the absence of EFR3A.

The fourth step is that increasing the localization and nanoclustering of KRAS on the plasma membrane enhances oncogenic signaling to promote tumorigenesis. Here we show that the that loss of EFR3A, EFR3B, or PI4KA all reduce P-ERK and P-AKT levels, as assessed by immunoblot, effects rescued by tethering KRAS$^{G12V}$ or PI4KA to the plasma membrane. Moreover, we show that loss of EFR3A, EFR3B, or PI4KA all reduce short-term (Titer-Glo analysis) and long-term (colony formation) 2D-transformed growth, as well as 3D growth-transformed growth (soft agar) in not only oncogenic KRAS-transformed HEK-HT cells, but also in multiple human pancreatic adenocarcinoma cell lines characterized by an endogenous mutant *KRAS* allele. Again, these phenotypes were rescued by tethering KRAS$^{G12V}$ or PI4KA to the plasma membrane. Furthermore, loss of EFR3A, EFR3B, or both reduced tumor growth of a human pancreatic adenocarcinoma cell line in vivo.

We recognize the possibility that loss of EFR3A, EFR3B, or PI4KA may reduce oncogenic signaling due to a general decrease in PI(4)P and PS levels at the plasma membrane, and in turn, KRAS localization and nanoclustering. However, the finding that EFR3A specifically binds KRAS in the active GTP-bound state and that *sgEFR3A* did not affect the growth of HEK-HT cells in the absence of oncogenic KRAS or in the KRAS wild-type pancreatic adenocarcinoma cell line BxPC3 supports, but admittedly does not prove, that it is the recruitment of EFR3A to KRAS that is a key step in the ability of EFR3A to promote oncogenic KRAS signaling. We also recognize the possibility that EFR3A may promote KRAS signaling through other proteins. However, the loss of EFR3A was phenocopied by the loss of PI4KA and rescued by tethering PI4KA to the plasma membrane, arguing against this possibility. Perhaps more likely, PI4KA may have effects beyond those described, especially given the role of phosphatidylinositol metabolism in membrane trafficking[25], there could be other mechanism fostering KRAS nanoclustering beyond a PI(4)P/PS exchange, or the elevation in PI(4)P recruits unknown proteins regulating RAS signaling.

We suggest that this unique signaling circuit has the potential to be therapeutically targeted. First, as noted above, we show that loss of EFR3A, EFR3B, both proteins, or PI4KA reduced onco-genic KRAS signaling, transformation, and/or tumorigenesis, both at the ectopic and endogenous level in multiple cell back-grounds. Interestingly, components of EFR3A signaling are upregulated in human pancreatic adenocarcinoma, but not other tested RAS-mutant (KRAS, NRAS, or HRAS) cancers. Whether this reflects a hyper-dependency on KRAS mutations in pancreatic adenocarcinoma, or some unique feature of pancreatic tumorigenesis remains to be elucidated. While EFR3A is not easily druggable, the kinase it associates with is. Indeed, an interplay of RAS and another class of PI4 kinases (PI5P4K) was uncovered as a product of the dual-inhibitory compound a131[67]. However, dosing PI4KA inhibitors can be challenging, as evidenced by testing different PI4K inhibitors developed for the treatment of Hepatitis C viral infection, that have exhibited a rather variable range of toxicity[63,64]. Here we show that low doses of the PI4KA-specific inhibitor C7 are synergistic with the clinical KRAS$^{G12C}$-specific inhibitor sotorasib, resulting in near complete ablation of the transformed growth of KRAS$^{G12C}$-mutant human cancer cell lines. Interestingly, as noted above, while components of the EFR3A complex were upregulated in KRAS mutation-positive pancreatic tumors, we nevertheless show synergy of C7 with sotorasib in KRAS$^{G12C}$-mutant human lung adenocarcinoma cell lines. Such a finding bodes well for the possible deployment of a PI4KA inhibitor to either enhance the antineoplastic activity of, or prevent resistance to KRAS$^{G12C}$ inhibitors in multiple cancer types. In conclusion, we uncover a component of the KRAS interactome that induces a unique positive-feedback circuit promoting KRAS oncogenesis that could be leveraged for the treatment of KRAS mutation-positive cancers.

# Methods

**Plasmids.** pWZL-Blast-myc-KRAS$^{G12V}$ was generated by PCR cloning myc epitope-tagged *KRAS$^{G12V}$* (4B splice variant) cDNA (Thermofisher Scientific) into the plasmid pWZL-Blast (a kind gift of Jay Morgenstern). pWZL-Blast-myc-KRAS$^{G12V}$(op) was generated in the same fashion except codon optimized. myc-BioID-pBabePuro-myc-BirA-KRAS$^{G12V}$ was previously described[42,68]. pcDNA3.1-FLAG-EFR3A was generated by cloning *FLAG-EFR3A* cDNAs (Genescript) into pcDNA3.1. pWZL-Neo-myr-myc-KRAS$^{G12V}$-SAAX and myr-FLAG-PI4KA were generated by introducing a myristoylation signal sequence (5′-ATGGGGTCTTC AAAATCTAAACCAAAGGACCCCAGCCAGCGCCGGCGCAGGATCCGAGG TTACCTT-3′)[55] at the N-terminus and a of *KRAS-SAAX* (ThermoFisher Scientific) and *PI4KA* cDNAs (Genescript) by PCR, which was then inserted into pWZL-Neo (Cell Biolabs). pET21a-His10-KRAS$^{WT}$ was generated by incorporating an N-terminal 10x His sequence in frame into the 5′ end of *KRAS* cDNA (ThermoFisher Scientific), which was then inserted into pET21a (EMD Millipore). pET21a-His10-KRAS$^{G12V}$ was generated in the same manner based on *KRAS$^{G12V}$* cDNA (Thermofisher Scientific). pGEX-6P2-GST-EFR3A was generated by PCR amplification of *EFR3A* cDNA (Genescript) and subcloned into pGEX-6P2 (GE Healthcare). LentiCRISPR-v2.0-Puro-sgEFR3A-1 and -2 were generated by cloning *sgEFR3A-1* (5′-TGAGAGGCTCATCCGTGACG-3′) or *sgEFR3A-2* (5′-GAAGTTT GCCAACATCGAGG-3′) into LentiCRISPR v2.0-Puro (Addgene vector 52961). LentiCRISPR-v2.0-Blast-sgEFR3B and sgPI4KA were generated by cloning *sgEFR3B-1* (5′-TGAGAGGCTCATCCGTGACG-3′) or *sgPI4KA* (5′-ATGTCTAAG AAAACCAACCG-3′) into LentiCRISPR-v2.0-Blast (a gift from Sina Ghaemma-ghani, University at Rochester). pCDH-GFP-KRAS$^{G12V}$-IRES-mCherry-CAAX, GFP-pCDH-GFP-LactC2/mCherry-CAAX, and pEGFP-SidM/mCherry-CAAX were as previously described[24]. plenti6.3-EFR3A-mCherry was generated by PCR amplification of *EFR3A-mCherry* fragment from pcDNA3.1 *EFR3A-mCherry* was then subcloned into plenti6.3. *EFR3A* cDNA (Genescript) was cloned into the N-terminus of the mCherry fragment of pcDNA3.1-mCherry. The *EFR3A-mCherry* fragment was then subcloned into plenti6.3 (Invitrogen). Similarly, plenti6.3-EFR3B-mCherry and plenti6.3-FAM126A-mCherry were generated by PCR amplification of *EFR3B-mCherry* and *FAM126A-mCherry* fragments from the respective pcDNA3.1-EFR3B-mCherry and -FAM126A-mCherry plasmids. pcDNA3.1-TTC7A-mCherry was generated by PCR amplification of *TTC7A* cDNA (Genescript) and cloned into the N-terminus of the mCherry fragment of pcDNA3.1-mCherry. All plasmids were confirmed by sequencing.

**Cell culture.** HEK-HT human embryonic kidney epithelial cells stably expressing *hTERT* and proteins encoded by the SV40 early region[46] were confirmed to be free of mycoplasma infections by the Duke Cell Culture Facility using MycoAlert PLUS test (Lonza). MDCK cells were previously described[24]. The human cell line 293T, the human pancreatic adenocarcinoma cell lines CFPAC-1, AsPC-1, HPAF-II, PANC-1, and BxPC3 and the human lung adenocarcinoma cell lines H358 and H2030 were purchased from Duke University Cell Culture Facility and similarly certified to be mycoplasma-free as above and also authenticated by STR analysis. All cell cultures were maintained at 5% CO$_2$ and cultured in media supplemented with 10% FBS (fetal bovine serum) and 1% pen/strep antibiotic mixture. HEK-HT and 293T cells were cultured in Dulbecco's Modified Eagle Medium (DMEM). MDCK cells were cultured in Minimum Essential Medium Eagle (MEM). CFPAC-1 cells were cultured in Iscove's Modified Dulbecco's Medium (IMDM). AsPC-1, BxPC3, H358, and H2030 cells were cultured in RPMI medium. HPAF-II cells were cultured in Minimum Eagle's Medium (MEM) supplemented with 1% NEAA and 1% sodium pyruvate. PANC-1 cells were cultured in DMEM. MiaPaCa-2 cells were cultured in DMEM supplemented with 2.5% equine serum. For EGF stimulation, BxPC3 cells were serum starved in 0.1% FBS for 24 h and then treated with EGF (50 ng/ml) for 5, 10, and 20 min.

**Cell lines.** Transient transfections were performed with 6 μg DNA and 18 μl of Fugene6 reagent (Promega Corporation) incubated in serum-free medium for 25 min according to the manufacturer's protocol. 48 h later the cells were analyzed. For lentivirus infection, 293T cells were seeded onto 10 cm plates and 24 h later transfected with a mixture containing 0.3 μg VSVg, 3 μg psPAX2, 3 μg DNA and 18 μl Fugene6 in OptiMEM. The following day, the medium was exchanged with DMEM medium containing 30% FBS. 48 h later, the virus was filtered through a 0.45 μm filter and then frozen at −80 °C until further use. 100 μl of the virus was used to transduce 20,000 cells seeded onto 6-well plates and selected on antibiotic medium for 7 days before being analyzed. Cell lines for evaluating the loss of EFR3A/B on RAS signaling were created by stably transducing KRAS$^{G12V}$-transformed HEK-HT, HPAF-II, PANC-1, AsPC-1, CFPAC-1, MiaPaCa-2, H358, and H2030 with lentiviruses encoding either vector (Cas9), *sgEFR3A-1*, *sgEFR3A-2*, *sgEFR3B*, or both *sgEFR3A-1* and *sgEFR3B-1*. For co-immunoprecipitation experiments of biotinylated proteins, 293T cells were infected with retroviruses derived from pBabePuro-myc-BirA-KRAS$^{G12V}$, -KRAS$^{WT}$, -or -KRAS$^{S17N}$, after which stable cells were transiently transfected with pcDNA3.1-FLAG-EFR3A plasmid[68]. For visualization of KRAS, HEK-HT cells stably expressing vector (Cas9) or *sgEFR3A-1* and *sgEFR3B-1* were transduced with lentiviruses encoding GFP-KRAS$^{G12V}$-IRES-mCherry-CAAX. Cells for membrane fractionation studies were generated by stably transducing KRAS$^{G12V}$-transformed HEK-HT, CFPAC-1, and HPAF-II cells with lentiviruses encoding *sgEFR3A-1*, or both *sgEFR3A-1* and

*sgEFR3B-1.* For rescue experiments, KRAS$^{G12V}$-transformed HEK-HT or HPAF-II cells were infected with lentiviruses encoding *sgEFR3A-1* and *sgEFR3B-1*, after which stably infected cells were infected with retrovirus encoding either pWZL-Neo-myr-myc-KRAS$^{G12V}$-SAAX or pWZL-Neo-myr-FLAG-PI4KA.

**Cell proliferation.** $2 \times 10^5$ cells from the indicated cell lines per well were seeded in triplicates in 6-well plates. 16 to 18 h later the cells were washed twice in PBS and refed with media supplemented with 0.5% FBS. Five days later the cells were stained with crystal violet using standard procedures[42].

**Soft agar growth.** Wells of a 6-well plate were seeded with $1.0 \times 10^6$ of HEK-HT cells stably infected with a retrovirus derived from pWZL-Blast-myc-KRAS$^{G12V}$(op) or $2.0 \times 10^5$ of the indicated pancreatic adenocarcinoma cells with the indicated perturbations in 1 ml of 0.3% bacto-agar (BD Bioscience) containing DMEM or RPMI solution. The bottom layer was prepared in 2 ml of 0.6% bacto-agar. Cells were fed with 500 μl of medium every three days. Colonies >20 cells in number were counted after 28 days in the case of HEK-HT cells. Colonies >50 cells in number were counted after 21 days in the case of pancreatic adenocarcinoma cells.

**Xenograft tumor assay.** $1 \times 10^7$ of the indicated HPAF-II cells were resuspended in 100 μl of 1:1 PBS/matrigel and subcutaneously injected into flanks of six-week-old female SCID/beige mice (Charles River). The injection site was measured twice weekly until tumors were palpable (~200 mm$^3$), after which measurements were increased to daily until reaching a maximum tumor volume (1,500 mm$^3$). All studies with vertebrate animals were performed under a protocol approved by Duke IACUC.

**Immunoblotting.** Cells were lysed in the lysis buffer containing 1% NP-40, 100 mM Tris HCl pH 8.0, 50 mM NaCl, 1 mM EDTA, 1 mM PMSF, and protease inhibitors (Roche). Protein extracts were separated on 10-15% gradient SDS-PAGE gels (BioRAD) and transferred onto PVDF membranes using BioRAD Turbo blot system. Membranes were blocked in 5% dry milk or BSA (bovine serum albumin), incubated with the primary (see below) and then appropriate secondary antibodies, before being processed for signal detection using enhanced chemiluminescence reagent (ThermoFisher Scientific). Images were digitally retrieved using Chemi Doc Imager (BioRad). Primary antibodies used detected EFR3A (ThermoFisher Scientific #PA5-54694; diluted 1:100), EFR3B (ThermoFisher Scientific #PA5-107118; diluted 1:250), PI4KA (ThermoFisher Scientific #PA5-28570; diluted 1:50) myc (Cell Signaling #2276; diluted 1:1000), β-actin (Cell Signaling #3700; diluted 1:5000), FLAG (Sigma #F3165; diluted 1:1000), KRAS (Sigma #WH0003845M1; diluted 1:1000), P-ERK1/2$^{T202,Y204}$ (Cell Signaling #9101; diluted 1:1000), P-AKT$^{308}$ (Cell Signaling #9271; 1:500), AKT (Cell Signaling #9272; diluted 1:1000), ERK (Cell Signaling #9102; diluted 1:1000), BRAF (Cell Signaling #14814; diluted 1:5000), CRAF (Cell signaling #53745; diluted 1:500) and P-CRAF (Cell Signaling #9427; diluted1:500), GST (Santacruz #sc-138; diluted at 1:1000), Membrane fraction WB cocktail (Abcam #: diluted 1:1000) and Strep-HRP (ThermoFisher Scientific # SA10001; diluted 1:20000). Secondary antibodies used were goat anti-rabbit IgG (H + L) HRP (Thermo Fisher Scientific, #65-6120, 1:3000) or goat anti-mouse IgG (H + L) HRP (Life Technologies, #G21040, 1:5000). Different exposure times were used to optimize detection of each protein.

**RBD affinity capture.** Affinity capture of RAS-GTP by the RBD pull-down method was performed as per the manufacturer's instructions (Cell Signaling #8821). Cell lysates were incubated with 10 μl of glutathione beads and 100 μg GST-RBD for 1 h at 4 °C with end-to-end rotation. The beads were washed five times with manufacturer-supplied wash buffer. Bound proteins were eluted in SDS loading buffer heated 95 °C for 5 min and then resolved by SDS-PAGE. GTP-bound KRAS was detected by immunoblot with a KRAS4B-specific antibody. The same assay was performed with recombinant proteins, except recombinant KRAS proteins were first loaded with either GTPγS or GDP. Briefly, >0.5 mg of the recombinant proteins were incubated with 20 μl of 0.5 M EDTA and 2 mM of GTPγS or GDP for 45 min at room temperature with gentle rotation. The reactions were stopped by the addition of 60 μl of 1 M MgCl$_2$.

**Protein purification and size-exclusion chromatography.** BL21 Rosetta cells were transformed with expression vectors encoding full-length His$_{10}$-KRAS$^{WT}$, full-length His$_{10}$-KRAS$^{G12V}$, and GST-EFR3A (108-821). We note that full-length EFR3A could not be expressed in bacteria. Thus, based on previous studies demonstrating that deletion of the N-terminal membrane targeting region permitted the yeast version to be expressed, we generated a similar N-terminal deletion protein. Transformed cells were grown in LB media to an OD$_{600}$ of 0.6, then induced with 0.5 mM isopropyl β-D-1-thiogalactopyranoside at 16 °C and harvested 16 h later. Cell pellets were resuspended in Buffer A (20 mM HEPES pH 7.5, 200 mM NaCl, and 2 mM β-mercaptoethanol) supplemented with 1 mM PMSF and protease inhibitor cocktail (Roche). The cells were lysed using sonication (10 seconds; 30 pulses) and clarified by centrifugation. With regards to His$_{10}$-KRAS$^{WT}$ and His10-KRAS$^{G12V}$ proteins, the clarified lysates were passed over Ni-

NTA resin that was pre-equilibrated with Buffer B (20 mM HEPES pH 7.5, 150 mM NaCl, 2 mM β-mercaptoethanol, and 1 mM PMSF) containing 5 mM imidazole. The column was washed with Buffer B with increasing concentrations of imidazole and eluted with Buffer B containing 250 mM imidazole. With regards to GST-EFR3A protein, the clarified lysate was purified using Glutathione sepharose 4B resin (GE Healthcare) pre-equilibrated with Buffer A. The column was washed with ten column volume of Buffer A and eluted with Buffer A containing 10 mM reduced glutathione. Protein concentrations were measured using NanoDrop 1000 (Thermo Fisher Scientific). The purified proteins were aliquoted, flash frozen in liquid nitrogen, and stored at −80 °C. Purified proteins were subjected to size-exclusion chromatography (SEC) using Superose® 6 Increase 10/300 GL (Cytiva) pre-equilibrated with Buffer C (20 mM HEPES pH 7.5, 150 mM NaCl, 1 mM DTT) at a flowrate of 0.2 ml per minute. To test for an EFR3A-KRAS interaction, 16 μM SEC-purified $His_{10}$-$KRAS^{G12V}$ was loaded with 2 mM GTPγS and incubated with 12.8 μM SEC-purified EFR3A for 1 h at 4 °C with end-to-end rotation. The mixture was centrifuged for 10 min at 12,000 $g$ and subjected to SEC exactly as above. Fractions from individual GST-EFR3A, $His_{10}$-$KRAS^{G12V}$, and GST-EFR3A + $His_{10}$-$KRAS^{G12V}$ runs were analyzed by immunoblot.

**Far western analysis.** Recombinant proteins were derived as above, except in one experiment (Fig. 4k) GST-EFR3A was purchased from a commercial vendor (Novus Biologicals, H00023167). 150 μg/ml of recombinant GST-EFR3A was dot blotted on a nitrocellulose membrane, which was then blocked with 5% BSA and incubated with 10 μg/ml recombinant $His_{10}$-$KRAS^{WT}$ and $His_{10}$-$KRAS^{G12V}$ in 5% BSA for 2 h at room temperature. Membranes were washed at least three times with 1X TBST containing 5% BSA at room temperature and then incubated with KRAS4B- and EFR3A-specific antibodies at 4 °C overnight. Membranes were washed at least three times with 1X TBST and incubated with secondary antibody for 1 h at room temperature followed by three additional washes with 1X TBST. Resultant blots were developed using standard chemiluminescence detection.

**in vitro coimmunoprecipitation.** 50 μg of purified recombinant $His_{10}$-$KRAS^{G12V}$ was incubated with 150 μg of purified recombinant GST-EFR3A for 1.5 h at 4 °C in immunoprecipitation (IP) Buffer (50 mM Tris HCl, 150 mM NaCl, 1% Triton X-100 and a tablet of protease inhibitor cocktail (Roche)). 25 μl of precleared Protein G Dynabeads (ThermoFisher Scientific) bound to either a KRAS-specific antibody or IgG isotype control was added and the mixture was incubated for another 1.5 h at 4 °C, after which the beads were washed three times in the above mentioned IP Buffer at room temperature. As a negative control, IgG from the same animal species was used. Bound proteins were eluted in SDS loading buffer heated at 95 °C for 5 min.

**BRAF competition assay.** Recombinant BRAF protein (Abcam #ab55690) was incubated at increasing concentrations (0 μM, 0.25 μM, 0.5 μM, and 1 μM) with 150 μg purified recombinant EFR3A and 50 μg purified recombinant KRAS pre-loaded as above with GTPγS for 1.5 h at 4 °C in immunoprecipitation buffer (50 mM Tris HCl, 150 mM NaCl, 1% Triton X-100 and a tablet of protease inhibitor cocktail (Roche)). The mixture was then subjected to in vitro co-immunoprecipitation as described above.

**Co-immunoprecipitation of endogenous EFR3A with ectopic $KRAS^{G12D}$.** pcDNA3.1-FLAG-$KRAS^{G12D}$ plasmid was transiently transfected in $1 \times 10^7$ of 293T cells as above. 48 h later, cells were washed twice with ice-cold 1X PBS and lysed in immunoprecipitation buffer (50 mM Tris HCl, 150 mM NaCl, 1% Triton X-100 and a tablet of protease inhibitor cocktail (Roche)). Cell lysates were incubated with anti-FLAG-M2 resin (Sigma #A2220) overnight at 4 °C by end-to-end rotation. The next day, the beads were washed three times in the above mentioned IP buffer at 4 °C. As a negative control, IgG from the same animal species was used. Bound proteins were eluted in SDS loading buffer by heating at 95 °C for 5 min prior to immunoblotting as described above.

**Live-cell confocal microscopy.** HEK-HT stably infected with lentiviruses derived from LentiCRISPR-v2.0-Puro-sgEFR3A-1 and -sgEFR3B were stably infected with lentivirus derived pCDH-GFP-$KRAS^{G12V}$-IRES-mCherry-CAAX. 72 h later the cells were plated on 35 mm culture plates. 24 h later cells were placed in a humidifying chamber controlled at 37 °C with 5% $CO_2$ and live cell images collected using Andor Dragonfly spinning disk confocal microscope using 63 × 1.20 water immersion UPlan S-APO objective using an Andor iXon 897 EM-CCD camera.

**Membrane fractionation.** $2 \times 10^7$ of the indicated derived CFPAC-1 or HEK-HT cells were grown in a 15 cm dish. Cell membrane and cytoplasmic fractions were then isolated using ProteoExtract Native Membrane Protein Extraction Kit (Millipore Sigma #444810) following the manufacturer's protocol. Briefly, the cell monolayer was washed twice with prechilled wash buffer at room temperature. 1 ml Extraction Buffer I containing protease inhibitor cocktail was then added directly onto the cell culture. The cell culture plates were rocked gently at 4 °C for 10 min. Lysate were collected and centrifuged at 16,000 $g$ for 15 min at 4 °C. The resulting supernatant was aliquoted as cytoplasmic/soluble fraction while the pellet

was resuspended in 500 μl prechilled Extraction Buffer II containing protease inhibitor cocktail and incubated for 30 min at 4 °C with end-to-end rotation. The mixture was then centrifuged at 16,000 $g$ for 15 min at 4 °C. This resulting supernatant fraction was aliquoted as the membrane-enriched fraction. The respective fractions were then immediately processed for downstream immunoblot analysis.

**Spatial distribution and clustering of KRAS by transmission electron microscopy.** Univariate K-Function Analysis for analyzing the distribution of one species on the plasma membrane. Basal plasma membrane sheets of parental and EFR3A/Bknockout HEK-HT and MDCK cells were prepared, fixed with 4% paraformaldehyde, 0.1% glutaraldehyde, and labeled with 4.5 nm gold directly conjugated to anti-GFP antibody[11,23,69]. Images were obtained with a transmission electron microscope at 100,000x magnification and intact 1μm² areas were identified with ImageJ. Univariate K function analysis was performed and bootstrap tests were used to examine for statistical differences between replicated point patterns. Bootstrap tests were used to examine for statistical differences between replicated point patterns[70,71] and statistical significance was evaluated against 1000 bootstrap samples. Lmax: Each L(r)-r curve is a mean univariate K-function of 30 replicates of each treatment group. Curves above the 99% confidence interval (=1) indicate significant clustering. Lmax is used as a summary statistic to visualize the extent of clustering and is equal to the maximum observed value of the mean L(r)-r function of each group. Bivariate K-Function Analysis to quantify the extent of co-clustering between two different species on the plasma membrane. 293T cells were seeded onto fibronectin-coated cover slips. 24 h later the cells were transiently co-transfected with mGFP- and mCherry- or mRFP-tagged proteins of interest. After 48 h, plasma membrane sheets from these cells were attached to EM grids, fixed with 4% paraformaldehyde, 0.1% glutaraldehyde, and labeled with 6 nm gold directly conjugated to anti-GFP antibody and 2 nm gold conjugated to anti-RFP antibody[11,23]. Images were obtained with a transmission electron microscope at 100,000x magnification and intact 1 μm² areas were identified with ImageJ. Bivariate K function analysis was performed and bootstrap tests were used to examine for statistical differences between replicated point patterns[70,71] and statistical significance was evaluated against 1000 bootstrap samples. LBI: Each Lbiv(r)-r curve is a mean bivariate K-function of 30 replicates of each treatment group. Curves above the 95% confidence interval (=1) indicate significant co-clustering. LBI is used as a summary statistic to visualize the extent of co-clustering and is equal to the integral of the mean Lbiv(r)-r function of each group. LBI values above the 95% confidence interval (=100) indicate significant co-clustering.

**Drug treatment and cell viability assay.** The $RAS^{G12C}$ inhibitor sotorasib (AMG510) was purchased from Medchemexpress. The PI4KA inhibitor C7 was purchased from Ximbio. Both drugs were dissolved in DMSO. 20,000 cells were seeded in triplicate in 96 well plates. After 24 h, media was exchanged with fresh media containing increasing concentration (0, 5, 10, and 20 nM) of the indicated drugs. DMSO was used for vehicle-treated samples. After 72 h of incubation, viability was measured using Cell Titer Glo assay (Promega Corporation) following manufacturer's protocol.

**Bioinformatic and statistical analysis.** The 3-dimensional RAS CRISPR enrichment scatter plot was generated using Graphing Calculator 3D. cBioportal PanCan TCGA dataset was used to identify genomic alterations in RAS-mutant cancers. cBioportal (cbioportal.org) pancreatic adenocarcinoma dataset was used to generate survival plots[72,73]. EFR3A gene expression profiles in normal and tumor samples were analyzed in GEPIA portal (gepia.cancer-pku.cn) using the PAAD (Pancreatic Adenocarcinoma), LUAD (Lung Adenocarcinoma), COAD (Colorectal Adenocarcinoma), BLCA (Bladder Cancer) and HNSC (Head and Neck Squamous Cell Carcinoma) dataset. The same dataset was analyzed for KRAS mutations status and compared to EFR3A expression levels. Survival plots depicting correlation between EFR3A, EFR3B, FAM126A, PI4KA, and TTC7A expression and pancreatic cancer patient samples survival outcome was determined using XENA TCGA pancreatic adenocarcinoma dataset (xenabrowser.net). Two-way t-tests were performed using GraphPad Prism (GraphPad Software), and $P$ values are indicated in the respective graphs. Asterisks are also shown in the respective graphs, $*p < 0.05$, $**p < 0.01$, $***p < 0.001$ and $****p < 0.0001$. Bioinformatic statistics are indicated in the respective figure legend descriptions. All cellular experiments were repeated at least three times. The in vivo experiments contained 5 mice for each treatment group and the sample size was chosen to reflect a difference in means of 20% with a power of 90%. Synergy plots and scores were calculated using Synergy Finder (synergyfinder.fimm.fi).

**Reporting summary.** Further information on research design is available in the Nature Research Reporting Summary linked to this article.

## Data availability
All data have been analyzed using publicly available datasets that can be downloaded at cbioportal database cbioportal.org/datasets as Pancreatic adenocarcinoma [ICGC, Nature 2012; QCMG, Nature 2016; TCGA, Firehose Legacy; TCGA, PanCancer Atlas and for all

other cancers, TCGA PanCancer Atlas, datasets for XENA are available at XENA database xenabrowser.net/datapages/ as TCGA Lung Adenocarcinoma, TCGA Bladder Cancer, TCGA Pancreatic Cancer, TCGA Colon Cancer, TCGA Head and Neck Cancer, TCGA Target GTEx) and datasets utilized in GEPIA analysis at gepia.cancer-pku.cn are available as TCGA Pancreatic Cancer dataset at the XENA database XENAbrowser.net/datapages). Source data are provided with this paper. All the other data are available within the article and its supplementary information. Source data are provided with this paper.

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

## Acknowledgements

This work was supported by the National Cancer Institute (K99CA248495 to H.A., R01CA123031 and P01CA203657 to C.M.C.), the Cancer Prevention and Research Institute of Texas (RP200047 to J.F.H.), the Andrew Sowell-Wade Huggins Fellowship/ Professorship in Cancer Research (W.E.K. and J.F.H.), and the NIH Duke Clinical and Translational Science Award (UL1TR002553 through a subaward to P.Z.). We thank members of the Counter laboratory and Dr. Aaron Hobbs for helpful discussions and the Shared Resources of the Duke Cancer Institute for technical support (supported by P30CA014236 from the National Cancer Institute).

## Author contributions

H.A., W.E.K., and S.K. performed experiments. H.A., W.E.K., S.K., P.Z., J.F.H., and C.M.C. contributed to the conception of the study, experimental design, data interpretation, and writing the manuscript.

## Competing interests

The authors declare no competing interests.
