## [Peer Review File · Nature Communications]

REVIEWER COMMENTS

Reviewer #1 (Remarks to the Author):

The manuscript by Adhikari et al describes the discovery of the role of PI4KA signalling in mediating oncogenic Ras signalling. The authors used a BioID based approach to identify proteins in close contact to oncogenic Ras, and pulled out the PI4KA adaptor protein EFR3A. They postulate that oncogenic Ras can promote EFR3A driven PI4KA activity, leading to increased PI4P and PS concentration (through the PI4P mediated increase in PS levels). As PI4P and PS play a role in promoting nanoclusters that Ras localises to this creates a reinforcing signal. Overall, this is an interesting manuscript that provides unique insight into how oncogenic Ras is regulated at the plasma membrane.

I do have a number of major revisions that the authors would need to address before the manuscript is ready for publication. I am not convinced that the data as presented is strongly supporting a direct complex, and without clear determination of a complex, the conclusions of the manuscript are not justified.

There are also a number of typos, and the inclusion of line numbering would have simplified pointing these out. A general read over of the document for clarity would be useful.

Main concerns.

1. One of the main hypothesis from the authors is that KRas (and other isoforms) are forming a direct complex with EFR3A at the plasma membrane. This currently is the weakest part of the proposed model.

It seems that there is a chance that the BioID approach is also identifying proteins that may congregate at the same membrane nano clusters. Both EFR3A and Ras isoforms will be lipidated and may be expected to exist in the same membrane clusters through lipidation (not direct interaction). The putative interaction between EFR3A and HRas/KRas was previously identified in data from Kovalski et al Mol Cell 2019, however this interaction was not fully explored.

The data as currently presented needs to be more fully explained and expanded to justify the proposed direct interaction.

There currently is no methodological details on how the recombinant proteins are purified in Fig 3. The pET vectors are mentioned, so I am assuming this is from bacteria? Are the Ras proteins loaded with nucleotide? What is the loading state of the WT and G12 mutant? Neither of the proteins produced this way are going to be lipidated, and so will be lacking a membrane as in the cell. This is required for strong interaction for many Ras binding partner (PI3K as an example). Details on the purity and purification procedure is needed to prove this is not just aggregate protein co-purifying as a large aggregate.

Specifically showing SDS-page of recombinant protein, and gel filtration traces showing monodisperse stable proteins. Ideally a biophysical measurement of some time could be utilised as well. Competing off EFR3A binding with a known KRas binding partner would be a plus.

Finally the data as presented in Fig 3 across all experiments currently does not allow for any idea of the repeatability of the interaction. This would need to be presented with quantification and replicates, ideally with data presented as scatterplots with each replicate plotted.

2. Knockdown of EFR3A leads to greatly decreased Akt and Erk signalling. The question is how much of this is driven through phosphoinositide substrate depletion. It appears that there is a possibility that the data here is consistent with Ras signalling amplification downstream of EFR3A being completely independent of a direct complex.

Loss of PI4KA activity will lead to decreased PI4,5P2, which would decrease PI3K activation, which could explain loss of Akt phosphorylation. The loss of PI4P also would lead to decreased PS, which

would be expected to lead to decreased Ras activation through decreased SOS activation of Ras. This all would lead to decreased clustering of Ras. This could occur completely separate of the complex. More detailed description of how PS controls Ras activation in the discussion would be useful.

3. Inhibition of PI4KA is acutely toxic (far more than inhibitors of almost all other PI kinases). This was a long standing target of pharma for HepC treatment, but was essentially abandoned due to severe toxicity (see Bojjireddy et al JBC 2014). This needs to be more clearly addressed in the discussion.

4. Did the IQGAP protein get identified in the BioID screen? Data from the Anderson group proposed (Choi et al 2016) that this protein can scaffold PI4KA, PIPKI, and PI3K, which might potentially explain the kinases and binding partners identified.

Minor

Please address typos throughout text.

Reviewer #2 (Remarks to the Author):

In this manuscript, Adhikari et al. seek to analyze the role of EFR3-PI4KA signaling axis in oncogenic features driven by RAS, particularly mutant KRAS. By using BirA-mediated proximity labeling, they identify EFR3A as a protein interacting with active form of RAS. Further assays indicate that EFR3A mediates oncogenic KRAS malignancies, especially pancreatic cancer, through affecting the membrane localization of RAS protein. Such mechanism can provide as a therapeutic target to overcome mutant KRAS-driven cancers.

While in general the data support the conclusions drawn, there are some aspects of this study, which would benefit from additional clarification or investigation to further increase the relevance.

1. The authors used entire Figure 1 to emphasize the clinical importance of EFR3A in human cancers (normal vs tumoral pancreatic tissues). As they indicate the specific role of EFR3A in Kras mutant cancer cells, it might be more meaningful to include more types of cancers, such as lung and colon cancers where KRAS also serves as a predominant onco-protein. The addition of those cancer types (KRAS mutation rate is 35% and 50% respectively) will also allow the authors to study EFR3A expression level in the contents of wt v.s. mt KRAS.

2. Some data in this manuscript indicate that EFR3A and B have a collaborative role in mediating canonical RAS signaling pathways (Figure 2C). While EFR3A and B share similar biochemical structure, it is unclear in their study whether EFR3A and B are functionally interchangeable in KRRAS mutant cancers. It might also be helpful to include the expression level of EFR3B in the analysis shown in Figure 1.

3. While this manuscript shows some sophisticated biochemical assays, I find the functional assays to indicate cellular malignancies (i.e. Fig 2b) disappointing. All the in vitro transformation/colony formation assays in Fig 2b, 4j, 5j and 5i lack replications and statistics. In addition, foci formation and/or the 3D growing ability will be better assays to indicate the malignant transformation in cells.

4. Minor point: Regarding the statistical analysis, it is important to note the difference between Mean-/+SEM (which is used in this manuscript) and Mean -/+ SD. The SEM is a measure of precision for an estimated population mean. SD is a measure of data variability around mean of a sample of population. Unlike SD, SEM is not a descriptive statistic and should not be used as such.

5. The authors suggest that EFR3A specifically interacts with active form of KRAS (Fig3). Such conclusion may be strengthened by examining the association of EFR3A and RAS-GTP (A simple RasRBD pull down assay will serve the purpose).

6. What is the function of EFR3A in cancer cells expressing wild type KRAS? A lot of lung cancer cell lines have wild type KRAS but harbor genetic alterations involved in RAS signaling pathways (such as EGFR mutation/application or BRAF mutation). It might strengthen this manuscript by examining the role of EFR3A in such lines.

7. Xue et al. identify an adaptive fitness mechanism that allows groups of cancer cells within a population to rapidly escape inhibition of KRAS-G12C inhibitor (Nature 2020). Briefly, the synthesis

of new KRAS(G12C) and its distribution between the active or inactive states modulates the divergent response. Here, the authors suggest that the combination of PI4KA inhibitor and KRAS-G12C inhibitor have synthetic inhibitory effects in tumor cell viability. Yet, the longest treatment last for 72 hours. It might strengthen the results by showing the effects in a longer treatment timepoint (colony formation assay, 10-14 days treatment). The obtained results can help authors and other Ras researchers to understand whether the inhibition of PI4K4 can overcome the potential resistance to KRAS-G12C inhibitors.

8. Minor point: Please run the pulldown results indicated in Fig 3a on the same blot with comparable film exposure time.

Reviewer #3 (Remarks to the Author):

In Adhikari et al. the authors set out to follow up on a RAS interactor, EFR3A, found in their previous study where they delineated the RAS interactome of G12V mutated RAS isoforms. In this previous study, using BioID proximity ligation the authors saw EFR3A enriched in both NRAS and KRAS pull-downs. Interestingly, EFR3A, from their CRISPR KO library of interactors, was among the most highly negatively enriched target genes in all three (H-N-KRAS) transformed RAS-G12V mutants.

In this study the authors look into the role of EFR3A in KRAS mutant cancer cells. EFR3A is amplified in a number of cancers and this frequency is among the highest in pancreatic adenocarcinomas. Given that KRAS is also highly altered in PDAC they focus primarily on EFR3A in KRAS driven PDAC cells. They demonstrate that EFR3A and potentially its isoform EFR3B are required for maintaining MAPK and PI3K signaling in several KRAS mutant PDAC cell lines. The authors further show that knockout of EFR3A affects KRAS localization to the plasma membrane and is likely the cause for decreased MAPK/PI3K signaling. Targeting mutant KRAS to the plasma membrane by fusing a myristylation sequence to KRAS can reverse the affects of EFR3A/B-KO by restoring both MAPK/PI3K signaling and cell growth. EFR3A forms a complex with PI4K and is necessary for PI4K recruitment to the membrane to phosphorylate and produce PI(4)P. Loss of PI4K association to the plasma membrane in EFR3A/B KO cells results in decreased PI(4)P and PS levels at the membrane. Anchoring of PI4K to the membrane in EFR3A/B-KO cells restores PS levels, PI3K signaling and colony formation of KRAS mutant cells. Lastly, the authors combine a PI4K inhibitor with a RAS-G12C inhibitor to demonstrate a potential therapeutic target for KRAS mutant tumors.

Major points to address:

1. EFR3A is significantly amplified in PDAC, however, median patient survival based on EFR3A alteration does not seem significant. Does alteration represented here refer only to amplification? Similarly, the survival probability in patients comparing high vs. low EFR3A is not significant? (Fig 1f). Is EFR3A amplification observed in cancers that are not predominantly RAS mutant (like HNSCC, bladder cancer etc)?
2. The authors correlate EFR3A mRNA expression in the tumor vs. normal samples. However, this does not stratify well relative to KRAS mutations suggesting EFR3A amplification is not a direct consequence of RAS mutations. Since, EFR3A was identified in a RASG12V screen, can the authors categorize patients based on codon mutations and check for EFR3A expression levels.
3. In Figure 2, the authors show that depletion of EFR3A results in reduction of pERK and a decrease in cell growth in vitro and in vivo. However, the effects on cell growth are relatively modest (i.e. Panc1 and AsPc1) and the in vivo study is not an intervention study but rather a tumor initiation study and can be confounded by take rates. Can the authors show that in a KRAS wildtype pancreatic cell line, depletion of EGFR3A has no effect? Do the lines they tested exhibit elevated EFR3A expression? Can the authors include xenograft study data for the individual depletion of EFR3A and EFR3B. Is there any PD readout from the study, to correlate tumor growth inhibition to pathway activity?
4. One of the major issues of the manuscript is evidence for whether the interaction between EFR3A/B with RAS truly exists. The KRAS-EFR3A interaction experiments are weak. First, the interaction is possibly so transient that none of the pull-downs can be done from endogenous protein or with just at least the prey at endogenous levels. The interactions are only seen when

both KRAS and EFR3A are ectopically over expressed in cells or when using proximity ligation. With proximity ligation experiments like BioID, the amount of biotin and incubation period can greatly affect the range of interactions. This seems to be an issue in interpreting a few of the results.

- a. Based on the BirA mediated proximity labeling and loss-of-function CRISPR screen, the authors identify EFR3A as a top interactor relative to BRAF. Along the lines, can the authors sparse the dataset based on strong interactors vs. weak interactors.
- b. Is BRAF used as a relative measure since it's a strong or a weak interactor of RAS oncoproteins?
- c. Given, CRAF and not BRAF is a top synthetic lethal hit in RAS driven cancers, it would be useful to map CRAF or determine EFR3A relative to CRAF.
- d. Since, BirA screen identifies transient interactions within a 10nM range, do other members of the EFR3 complex which include EFR3B, FAM126A/B, TTC7A/B transiently associated with RAS nanoclustering at the membrane.
5. In Fig 3B the authors pull-down for His10-KRAS-G12V, using a KRAS specific antibody, and probe for GST-EFR3A. In the top panel "CO-IP: EFR3A" the blot shows EFR3A comes down in the KRAS Ab IP, but not in the mock IP with IgG.
 - a. Is this interaction dependent on KRAS being loaded with GTP or GDP? They compare WT versus mutant KRAS interaction in Figure 3D but never specifically load KRAS with GDP or GTP to show specifically that the interaction is dependent on the active or inactive state of KRAS.
 - b. Can the authors include additional controls here showing that the interaction is specific? Can the authors also conduct a reverse IP? Can they determine the dissociation constant for the interaction? Additional controls with the recombinant proteins specifically would go a long way in convincing the reader that this interaction is specific.
 - c. Given that the cells used in Fig3B express GST-EFR3A can the authors do the reverse pull-down and IP for EFR3A and probe for KRAS as an alternate way to demonstrate an interaction?
 - d. In Fig 3D increasing amounts of GST-EFR3A is added to the membrane for the dot blot and yet in the bottom panel for the loading control it doesn't seem like there are increasing amounts of EFR3A. Can the authors account for this discrepancy?
 - e. In the IP using proximity ligation, Fig 3C, EFR3A is biotinylated only in the BirA-KRAS-G12V expressing cell and not KRAS-WT or dominant negative KRAS-S17N suggesting EFR3A associates primarily with active KRAS. What are the RAS-GTP levels in these proximity ligation experiments? Have the authors considered an EGF or growth factor stim experiment in KRAS-WT to see if this increases EFR3A association.
6. The authors argue that KRAS mutant PDAC cells are dependent on EFR3A, likely due to EFR3A's role in maintaining KRAS localization to the plasma membrane. However, all these effects of EFR3A KO can be explained by EFR3A recruiting PI4KA to the membrane to regulate PI(4)P and PS levels. Thus, loss of EFR3A could broadly impact localization of many membrane bound proteins that includes RAS. As EFR3A can already anchor to the plasma membrane through its N-terminal cysteine rich region the authors have not provided evidence for a requirement of KRAS for EFR3A function.
 - a. Given the authors demonstrate that recombinant EFR3A interacts with recombinant mutant KRAS, this suggests that membrane localization may not be required for the interacting regions. Can the authors narrow down potential interaction sequences with truncation mutants of EFR3A as it lacks an RBD domain.
7. Since the authors claim an epistatic relationship with EFA3A/3B and PI4KA, does PI4KA depletion affect ERK signaling? (Fig 5h). The blots are unclear and need to be repeated? Please include a pERK/ERK and pCRAF/total CRAF blots as well

There is also confusion with the writing in the manuscript for figure legends. Figure 3 legend has several errors and do not match with the figure itself.

1. In the figure legend it states that Fig 3D for the dot blot "...probed with increasing amounts of recombinant GST-EFR3A followed by immunoblot with an anti-FLAG antibody." Which protein contains the FLAG epitope? In Fig 3D the panels indicate anti-EFR3A being used and not FLAG or GST antibodies.
2. In figure legend 3B it's written as "Levels of oncogenic KRAS or HRAS associating with EFR3A..." but in this case where is HRAS or is this a typo?
3. Fig 3B also states "immunoprecipitating (IP) recombinant His10-KRAS-G12V by virtue of His10-epitope tagged..." doesn't that mean IP of the His10 tag, thus a Ni pull-down? Yet the figure is showing a KRAS Ab pull-down.

Minor concerns:

1. If EFR3B is epistatic to EFR3A is EFR3B upregulated as well in PDACs or other cancers?
2. Fig5H, what about pERK levels?
3. Does the PI4Ki inhibitor alone alter RAS association at the membrane? What about effects on MAPK and PI3K signaling?
4. Fig 5J labels are cut partially off

REVIEWER 1 COMMENTS

Comment 1: “One of the main hypothesis from the authors is that KRas (and other isoforms) are forming a direct complex with EFR3A at the plasma membrane. This currently is the weakest part of the proposed model. It seems that there is a chance that the BioID approach is also identifying proteins that may congregate at the same membrane nano clusters. Both EFR3A and Ras isoforms will be lipidated and may be expected to exist in the same membrane clusters through lipidation (not direct interaction). The putative interaction between EFR3A and HRas/KRas was previously identified in data from Kovalski et al Mol Cell 2019, however this interaction was not fully explored. The data as currently presented needs to be more fully explained and expanded to justify the proposed direct interaction. There currently is no methodological details on how the recombinant proteins are purified in Fig 3. The pET vectors are mentioned, so I am assuming this is from bacteria? Are the Ras proteins loaded with nucleotide? What is the loading state of the WT and G12 mutant? Neither of the proteins produced this way are going to be lipidated, and so will be lacking a membrane as in the cell. This is required for strong interaction for many Ras binding partner (PI3K as an example). Details on the purity and purification procedure is needed to prove this is not just aggregate protein co-purifying as a large aggregate. Specifically showing SDS-page of recombinant protein, and gel filtration traces showing monodisperse stable proteins. Ideally a biophysical measurement of some time could be utilised as well. Competing off EFR3A binding with a known KRas binding partner would be a plus. Finally the data as presented in Fig 3 across all experiments currently does not allow for any idea of the repeatability of the interaction. This would need to be presented with quantification and replicates, ideally with data presented as scatterplots with each replicate plotted.” We break this critique down into individual edits and experiments below for ease of discussion:

Comment 1a: “The putative interaction between EFR3A and HRas/KRas was previously identified in data from Kovalski et al Mol Cell 2019, however this interaction was not fully explored.”

Reply: As requested, we revised the text to clearly indicate that we previously reported that EFR3A was biotinylated BirA-KRAS^{G12V}, and further, that sgRNAs targeting this gene were negatively enriched in RAS-transformed cells (see Fig 2 and related text in Adhikari & Counter, *Nat Commun* **9**:3635, 2018). Since publishing these findings, two other groups (Bigenzahn et al., *Science* **362**:1171, 2018 and Kovalski et al., *Mol Cell* **73**:830, 2019) performed proximity labeling coupled to CRISPR/Cas9 loss-of-function analysis. Based on the reviewer's comments we mined these additional datasets and now add to the text that these groups similarly captured EFR3A/B or other components of this complex. Specifically, Kovalski et al. identified weak labeling of EFR3A and strong labeling of PI4KA, as well as significant depletion of EFR3A and PI4KA sgRNAs in some or all of the tested RAS-mutant cancer cell lines. Bigenzahn et al. found PI4KA, FAM126A, and TTC7B were labeled in their proximity experiments. One other study (Ritchie et al., *Cancer Genomics Proteomics* **14**:225, 2017) performed BirA-RAS proximity labeling, but did not report EFR3A in their list of top interactions, and they did not make their datasets available for analysis. We thank the reviewer ever so much for bringing up this point, as mining these independent databases revealed that EFR3A and other components were labeled by oncogenic BirA-RAS, and additionally, sgRNAs targeting EFR3A are negatively enriched, providing independent validation of this interaction.

Comment 1b: *“There currently is no methodological details on how the recombinant proteins are purified in Fig 3. The pET vectors are mentioned, so I am assuming this is from bacteria?”*

Reply: As requested, we now expand the materials and methods to describe the method of generating these proteins from bacteria. Specifically, BL21 Rosetta cells were transformed with full-length His₁₀-KRAS^{WT}, full-length His₁₀-KRAS^{G12V}, and GST-EFR3A (108-821). We note that full-length EFR3A could not be expressed in bacteria. Thus, based previous studies demonstrating that deletion of the N-terminal membrane targeting region permitted the yeast version to be expressed, we generated the similar N-terminal deletion protein. Transformed cells were grown in LB media to a OD₆₀₀ of 0.6, then induced with 0.5 mM isopropyl β-D-1-thiogalactopyranoside (IPTG) at 16 °C and harvested 16 hours later. Cell pellets were resuspended in Buffer A (20 mM HEPES pH 7.5, 200 mM NaCl, and 2 mM β-mercaptoethanol supplemented with 1 mM PMSF and protease inhibitor cocktail (Roche)). The cells were lysed using sonication (10 seconds; 30 pulses) and clarified by centrifugation. With regards to His₁₀-KRAS^{WT} and His₁₀-KRAS^{G12V} proteins, the clarified lysates were passed over Ni-NTA resin that was pre-equilibrated with Buffer B (20 mM HEPES pH 7.5, 150 mM NaCl, 2 mM β-mercaptoethanol, and 1 mM PMSF) containing 5 mM imidazole. The column was washed with Buffer B with increasing concentrations of imidazole and eluted with Buffer B containing 250 mM imidazole. With regards to GST-EFR3A protein, the clarified lysate was purified using Glutathione Sepharose 4B resin (GE Healthcare) pre-equilibrated with Buffer A. The column was washed with ten column volume of Buffer A and eluted with Buffer A containing 10 mM reduced glutathione. Protein concentrations were measured using NanoDrop 1000 (Thermo Fisher Scientific). The purified proteins were aliquoted, flash frozen in liquid nitrogen, and stored at -80 °C. Purified proteins were subjected to size-exclusion chromatography (SEC) using Superose® 6 Increase 10/300 GL (Cytiva) pre-equilibrated with Buffer C (20 mM HEPES pH 7.5, 150 mM NaCl, 1 mM DTT) at a flowrate of 0.2 ml per minute. To test for an EFR3A-KRAS interaction, 16 μM SEC-purified His₁₀-KRAS^{G12V} was pre-loaded with 2 mM GTPγS and incubated with 12.8 μM SEC-purified EFR3A for 1 hour at 4 °C with end-to-end rotation. The mixture was centrifuged for 10 minutes at 12,000g and subjected to SEC exactly as above. Fractions from individual GST-EFR3A, His₁₀-KRAS^{G12V}, and GST-EFR3A + His₁₀-KRAS^{G12V} runs were analyzed by immunoblot.

Comment 1c: *“Are the Ras proteins loaded with nucleotide?”*

Reply: As requested, we clarify that KRAS was loaded with nucleotide in **Fig 3g-j,o-s,u-w**.

Comment 1d: *“What is the loading state of the WT and G12 mutant?”*

Reply: As requested, we now note in the figure legend that in **Fig 3e,k** the GTP-loading status of KRAS was not determined. Based on this, however, we added subsequent experiments whereby we confirmed the GTP-loading status by an RBD pull-down assay (**Figs 3g, S4a**). Given this, we repeated this original experiment with wild-type and G12V active KRAS in the absence of no nucleotide, GDP, and non-hydrolyzable GTPγS, which confirmed that EFR3A co-immunoprecipitates only with GTP-loaded KRAS (**Fig 3g,h**). In addition, we pre-loaded recombinant KRAS^{G12V} with non-hydrolyzable GTPγS for the additional far western analysis (**Fig 3i,j**), size exclusion chromatography (**Fig 3o-r**), competition analysis (**Fig 3s**), and deletion analysis (**Fig 3u-w**).

Comment 1e: *“Details on the purity and purification procedure is needed to prove this is not just aggregate protein co-purifying as a large aggregate. Specifically showing SDS-page of recombinant protein”*

Reply: As requested, we now show the SDS-PAGE separation of KRAS protein for **Fig 3d** in the new supplementary figure **Fig S4d,e**.

Comment 1f: *“(include) gel filtration traces showing monodisperse stable proteins.”*

Reply: As requested, we show gel filtration traces showing monodisperse stable recombinant EFR3A and KRAS proteins used to demonstrate co-immunoprecipitation (**Fig 3e**) and also for the newly performed size exclusion chromatography experiment (**Fig 3m,o,q**).

Comment 1g: *“Ideally a biophysical measurement of some time could be utilised as well.”*

Reply: As requested, we employ the biophysical size exclusion chromatography to document the association of EFR3A with KRAS^{G12V} as a complex as shown by co-elution of both proteins at the same elution volume (**Fig 3 m-r**). This now validates the interaction of EFR3A with KRAS^{G12V} by the suggested biophysical approach, which again strengthens the conclusions.

Comment 1h: *“Competing off EFR3A binding with a known KRas binding partner would be a plus.”*

Reply: As requested, we performed the suggested competition experiment with recombinant BRAF, a known RAS effector, which we now show reduced co-immunoprecipitation of recombinant KRAS^{G12V} with recombinant ERF3A (**Fig 3s**). This argues against the association of EFR3A with KRAS^{G12V} being a product of aggregation.

Comment 1i: *“Finally the data as presented in Fig 3 across all experiments currently does not allow for any idea of the repeatability of the interaction. This would need to be presented with quantification and replicates, ideally with data presented as scatterplots with each replicate plotted.”*

Reply: As requested, we performed replicate experiments for **Fig 3a-d** and other requested experiments (now **Fig 3a-f,k,l**). The replicate experiments were quantitated and shown as scatter plots.

Comment 2: *“Knockdown of EFR3A leads to greatly decreased Akt and Erk signalling. The question is how much of this is driven through phosphoinositide substrate depletion. It appears that there is a possibility that the data here is consistent with Ras signalling amplification downstream of EFR3A being completely independent of a direct complex. Loss of PI4KA activity will lead to decreased PI4,5P2, which would decrease PI3K activation, which could explain loss of Akt phosphorylation. The loss of PI4P also would lead to decreased PS, which would be expected to lead to decreased Ras activation through decreased SOS activation of Ras. This all would lead to decreased clustering of Ras. This could occur completely separate of the complex. More detailed description of how PS controls Ras activation in the discussion would be useful.”*
We break this critique down into individual edits and experiments below for ease of discussion:

Comment 2a: *“...This could occur completely separate of the complex.”*

Reply: As requested, we now make a special note this important interpretation in the discussion. We note that the association of EFR3A with KRAS is GTP dependent, and that EFR3A sgRNA were not negatively enriched in HEK-HT cells lacking oncogenic RAS or in the

KRAS wild-type pancreatic cancer cell line BxPC-3. Thus, while we cannot rule out that the loss of EFR3A reduces KRAS signaling by altering the lipid content of the plasma membrane in a manner unrelated to its association with KRAS, the above data support a more targeted effect related to the specific association of active KRAS with EFR3A. In almost all respects, the model outlined by this reviewer is exactly how we envision EFR3A functioning in oncogenic KRAS signaling (also see below), the only deviation being that we suggest this effect is dependent upon EFR3A associating with active KRAS.

Comment 2b: *“More detailed description of how PS controls Ras activation in the discussion would be useful.”*

Reply: As requested, we now described in the discussion that depletion of PI(4)P on the plasma membrane as a consequence of loss of PI4KA activity will in turn deplete plasma membrane phosphatidylserine (PS) levels by depleting the PI4P gradient across the ORP5/8 lipid transporters as shown here and previously (Kattan et al, LSA 2019). As such loss of PI4KA activity (by direct knock down, EFRA3 depletion, or pharmacologic inhibition) phenocopies the effect of ORP5/8 knock down, which causes mis-localization of KRAS from the plasma membrane and abrogation of KRAS signal output. The molecular mechanism at play here is the exquisite binding specificity for PS that is “hardwired” into the structure of the KRAS C-terminal membrane anchor (Zhou et al, Cell 2017). The KRAS membrane anchor selectively interacts with asymmetric unsaturated PS (such as 16:0/18:1) to form nanoclusters on the plasma membrane that are required for KRAS signal output. Symmetric unsaturated PS species (such as 16:1/18:1) can support KRAS PM binding but not nanoclustering (Zhou et al, Cell 2017). Depleting the plasma membrane of all PS species therefore causes both loss of KRAS from the plasma membrane and de-clustering of KRAS that remains plasma membrane bound; both of these effects on KRAS plasma membrane interactions result in decreased KRAS signaling.

Comment 3: *“Inhibition of PI4KA is acutely toxic (far more than inhibitors of almost all other PI kinases). This was a long standing target of pharma for HepC treatment, but was essentially abandoned due to severe toxicity (see Bojjireddy et al JBC 2014). This needs to be more clearly addressed in the discussion.”*

Reply: As requested, we now cite this important study, noting that the PI4KA-specific inhibitor F1 developed for HepC treatment exhibited acute toxicity in mice. We also note that the HepC drug Simeprevir, which can inhibit PI4KA, increased the sensitivity to radiation therapy in breast and brain cancer xenograft models (Park et al., *Oncotarget* **8**:110392, 2017 and Kattan and Hancock, *Biochemical J* **477**:2892, 2020), and that our own studies using a low dose (IC₁₀) of the PI4KA-specific inhibitor C7 was synergistic with a similarly low dose (IC₁₀) of the KRAS^{G12C} inhibitor Sotorasib (e.g. new data now included as **Fig 6b,c**). We thus discuss the potential of using low doses in conjunction with RAS inhibitors to overcome the toxicity associated with PI4KA inhibitors.

Comment 4: *“Did the IQGAP protein get identified in the BioID screen? Data from the Anderson group proposed (Choi et al 2016) that this protein can scaffold PI4KA, PIPKI, and PI3K, which might potentially explain the kinases and binding partners identified.”*

Reply: As requested, we mined our BioID datasets, but did not observe biotinylation of IQGAP. We can include this point in the discussion if the reviewer feels it is an important point.

Minor note 1: *“Please address typos throughout text.”*

Reply: *As requested*, we have now addressed the typos throughout the text.

REVIEWER 2 COMMENTS

Comment 1: *“The authors used entire Figure 1 to emphasize the clinical importance of EFR3A in human cancers (normal vs tumoral pancreatic tissues). As they indicate the specific role of EFR3A in Kras mutant cancer cells, it might be more meaningful to include more types of cancers, such as lung and colon cancers where KRAS also serves as a predominant onco-protein. The addition of those cancer types (KRAS mutation rate is 35% and 50% respectively) will also allow the authors to study EFR3A expression level in the contents of wt v.s. mt KRAS.”*

Reply: As requested, we performed bioinformatic analyses of *EFR3A* levels across a wide spectrum of sequenced human cancers characterized by RAS mutations. We also compared *EFR3A* gene amplification in two examples of cancers characterized by oncogenic KRAS, versus NRAS, versus HRAS mutations, finding that *EFR3A* gene amplification was most prominent in KRAS-mutant cancers. As *EFR3A* was most commonly amplified in pancreatic cancer we focused on this disease. We now show that higher *EFR3A* expression and amplification tracks with the *KRAS* mutations status in pancreatic adenocarcinoma samples (**Fig 1f**). Extending this analysis to the other components of the *EFR3A* signaling complex, we now report an increase in the expression of *EFR3B* (**Fig 1h**), *PI4KA* (**Fig 1i**), *FAM126A* (**Fig 1j**), and *TTC7A* (**Fig 1k**) in pancreatic adenocarcinoma samples compared to matched normal tissue. This increase in the expression of components of the *EFR3A* signaling complex appears to favor pancreatic cancer, as these genes were not increased in four other RAS-mutant cancers we examined (**Fig S2c**). This bias may reflect a hyper-dependency on KRAS signaling in pancreatic cancer, as evident by the extremely high frequency of *KRAS* mutations (nearly 100%) in this disease. Thus, we solidify the original observation of *EFR3A* upregulation in pancreatic cancer to other components of the complex, and further, show that this increase tracks with the extreme bias of pancreatic cancer towards *KRAS* mutations. However, this does not discount the potential therapeutic utility of targeting the *EFR3A* signaling complex in other *KRAS*-mutant cancers, as we demonstrate that lung adenocarcinoma cell lines are highly sensitive to a PI4KA inhibitor in combination with a *KRAS*^{G12C} inhibitor.

Comment 2: *“Some data in this manuscript indicate that EFR3A and B have a collaborative role in mediating canonical RAS signaling pathways (Figure 2C). While EFR3A and B share similar biochemical structure, it is unclear in their study whether EFR3A and B are functionally interchangeable in KRRAS mutant cancers. It might also be helpful to include the expression level of EFR3B in the analysis shown in Figure1.”*

Reply: As requested, we analyzed the expression of *EFR3B* in a variety of cancer types, and consistent with our analysis of *EFR3A*, find *EFR3B* is expressed higher in pancreatic adenocarcinoma compared to normal matched tissue (**Fig 1h**), although not to the same extent as *EFR3A* (**Fig 1d**). We also note there that the combined loss of *EFR3A* and *EFR3B* is almost always more potent than the loss of either one alone in multiple assays.

Comment 3: *“While this manuscript shows some sophisticated biochemical assays, I find the functional assays to indicate cellular malignancies (i.e. Fig 2b) disappointing. All the in vitro transformation/colony formation assays in Fig 2b, 4j, 5j and 5i lack replications and statistics. In addition, foci formation and/or the 3D growing ability will be better assays to indicate the malignant transformation in cells.”* We break this critique down into individual edits and experiments below for ease of discussion:

Comment 3a: “While this manuscript shows some sophisticated biochemical assays, I find the functional assays to indicate cellular malignancies (i.e. Fig 2b) disappointing.”

Reply: As requested, we expanded this analysis to disrupt *EFR3B* as well as *EFR3A/B* and further provide images of higher resolution in the revised **Fig 2b**.

Comment 3b: “All the *in vitro* transformation/colony formation assays in Fig 2b, 4j, 5j and 5i lack replications and statistics.”

Reply: As requested, we now note in the figure legends that all transformation and colony formation assays were performed using three biological replicates assayed in triplicate. We also now include statistical comparisons for all transformation and colony formation assays.

Comment 3c: “in addition, foci formation and/or the 3D growing ability will be better assays to indicate the malignant transformation in cells.”

Reply: As requested, we performed the more rigorous 3D-transformed growth assay (colony formation in soft agar) and demonstrate that *EFR3A*, *EFR3B*, and/or *EFR3A/B* sgRNAs reduce the growth of KRAS-transformed HEK-HT cells (**Fig 2d**) and the KRAS-mutant human pancreatic cancer cell lines HPAF-II, PANC-1, AsPC-1, and CFPac-1, but less (or not) the KRAS mutation-negative pancreatic cancer cell line BxPC-3 (**Fig S3a**). Similarly, we demonstrate that the loss of 3D-growth upon knockout of the *EFR3A* and *EFR3B* genes is restored by expressing myr-KRAS^{G12V} (**Fig S7**). Finally, we similarly validate that *PI4KA* sgRNA reduces the 3D-transformed growth in soft agar of AsPC-1, PANC-1, and HPAF-II cells (**Fig S8a**), and further, that the loss of 3D-transformed growth upon knocking out *EFR3A* and *EFR3B* was rescued by expressing myr-PI4KA (**Fig S8b**). Thus, all major observations observed in 2D-transformed growth assays were recapitulated in 3D-transformed growth assays.

Comment 4: “Minor point: Regarding the statistical analysis, it is important to note the difference between Mean-/+SEM (which is used in this manuscript) and Mean -/+ SD. The SEM is a measure of precision for an estimated population mean. SD is a measure of data variability around mean of a sample of population. Unlike SD, SEM is not a descriptive statistic and should not be used as such”

Reply: As requested, we calculate and report SD values for all assays except TEM experiments in which SEM is desirable to portray the precision of the data.

Comment 5: “The authors suggest that *EFR3A* specifically interacts with active form of KRAS (Fig3). Such conclusion may be strengthened by examining the association of *EFR3A* and RAS-GTP (A simple RasRBD pull down assay will serve the purpose).”

Reply: As requested, we captured activated KRAS protein on RBD-conjugated beads from lysates derived from the homozygous KRAS^{G12D}-mutant human pancreatic cancer cell line AsPC-1, followed by immunoblot with an anti-*EFR3A* antibody, demonstrating that **endogenous** active KRAS associates with **endogenous** *EFR3A* in cells (**Fig 3z**). Also see further data supporting the interaction of *EFR3A* with active KRAS discussed in our response to reviewer 1, concerns 1d and 1h.

Comment 6: *“What is the function of EFR3A in cancer cells expressing wild type KRAS? A lot of lung cancer cell lines have wild type KRAS but harbor genetic alterations involved in RAS signaling pathways (such as EGFR mutation/application or BRAF mutation). It might strengthen this manuscript by examining the role of EFR3A in such lines”*

Reply: As requested, we tested the effect of loss of *EFR3A* or both *EFR3A* and *EFR3B* in the *KRAS* mutation-negative human cell line BxPC-3. We chose this pancreatic cancer cell line, as opposed to the suggestion of a lung cancer cell line, as the majority of our analysis is performed in pancreatic cancer cell lines, and BxPC-3 cells lack a *KRAS* mutation. We find little to no effect upon the loss of these genes on the level of phosphorylated (P) P-ERK and P-AKT (**Fig S3b**), 2D-transformed growth as assessed by the colony formation assay (**Fig S3c,d**), and 3D-transformed growth, as assessed by the soft agar assay (**Fig S3a**).

Comment 7: *“Xue et al. identify an adaptive fitness mechanism that allows groups of cancer cells within a population to rapidly escape inhibition of KRAS-G12C inhibitor (Nature 2020). Briefly, the synthesis of new KRAS(G12C) and its distribution between the active or inactive states modulates the divergent response. Here, the authors suggest that the combination of PI4KA inhibitor and KRAS-G12C inhibitor have synthetic inhibitory effects in tumor cell viability. Yet, the longest treatment last for 72 hours. It might strengthen the results by showing the effects in a longer treatment timepoint (colony formation assay, 10-14 days treatment). The obtained results can help authors and other Ras researchers to understand whether the inhibition of PI4K4 can overcome the potential resistance to KRAS-G12C inhibitors.”*

Reply: As requested, we tested and now show that long-term (12 days) treatment with RAS^{G12C} and PI4KA inhibitors at a low IC₁₀ results in near complete ablation of colony formation (**Fig 6b,c**). We thank the reviewer for this suggestion, as it further supports the clinical potential of PI4K inhibitors to counter resistance to the KRAS^{G12C} drug and possible other RAS inhibitors being developed.

Comment 8: *“Minor point: Please run the pulldown results indicated in Fig 3a on the same blot with comparable film exposure time.”*

Reply: As requested, we now included a new gel image with myc-BirA and myc-BirA-KRAS^{G12V} on the same blot and using the same exposure time (**Fig 3a,b**).

REVIEWER 3 COMMENTS

Comment 1: *“The EFR3A is significantly amplified in PDAC, however, median patient survival based on EFR3A alteration does not seem significant. Does alteration represented here refer only to amplification? Similarly, the survival probability in patients comparing high vs. low EFR3A is not significant? (Fig 1f). Is EFR3A amplification observed in cancers that are not predominantly RAS mutant (like HNSCC, bladder cancer etc)?”* We break this critique down into individual edits and experiments below for ease of discussion:

Comment 1a: *“...median patient survival based on EFR3A alteration does not seem significant.”*

Reply: *As requested*, we clarify here that the survival probability in **Fig 1c** has a p value of 0.01781, which is statistically significant ($P < 0.05$).

Comment 1b: *“Does alteration represented here refer only to amplification?”*

Reply: *As requested*, we now clarify that the plot in **Fig 1c** is for all genomic alterations, although please note that amplification accounts for 95% of the genomic alterations observed in these patients (see **Fig 1b**)

Comment 1c: *“Similarly, the survival probability in patients comparing high vs. low EFR3A is not significant?”*

Reply: *As requested*, we clarify here that the survival probability in **Fig 1g** has a p value of 0.01781, which is statistically significant ($P < 0.05$).

Comment 1d: *Is EFR3A amplification observed in cancers that are not predominantly RAS mutant (like HNSCC, bladder cancer etc)?*

Reply: *As requested*, we performed bioinformatic analyses of EFR3A mRNA expression across multiple cancer types. We find that of the various RAS-mutant cancers, EFR3A expression tracks best in pancreatic cancer (**Fig S2c**), while perhaps not to surprising, it is a bit of a mixed bag when it comes to other cancers. Interestingly, and discussed above in response to concern 1 of reviewer 2, genes encoding the other components of the EFR3A signaling complex were also more highly expressed in pancreatic cancer (**Fig 1h-k**).

Comment 2: *“The authors correlate EFR3A mRNA expression in the tumor vs. normal samples. However, this does not stratify well relative to KRAS mutations suggesting EFR3A amplification is not a direct consequence of RAS mutations. Since, EFR3A was identified in a RASG12V screen, can the authors categorize patients based on codon mutations and check for EFR3A expression levels.”*

Reply: *As requested*, we show a positive relationship between EFR3A mRNA expression (**Fig 1e**) and now EFR3A gene amplification (**Fig 1f**) in KRAS-mutant versus KRAS wild-type pancreatic cancer patients (**Fig 1e**). We also note in the revised text that this correlation does not extend to other KRAS-mutant cancers, perhaps pointing to a hyper-dependency on KRAS mutations in pancreatic cancer. In support, we find that expression of the other components of the EFR3A signaling complex are similarly elevated in tumor versus normal tissue from pancreatic cancer patients (**Fig 1h-k**).

Comment 3: *“In Figure 2, the authors show that depletion of EFR3A results in reduction of pERK and a decrease in cell growth in vitro and in vivo. However, the effects on cell growth are relatively modest (i.e. Panc1 and AsPc1) and the in vivo study is not an intervention study but rather a tumor initiation study and can be confounded by take rates. Can the authors show that in a KRAS wildtype pancreatic cell line, depletion of EGFR3A has no effect? Do the lines they tested exhibit elevated EFR3A expression? Can the authors include xenograft study data for the individual depletion of EFR3A and EFR3B. Is there any PD readout from the study, to correlate tumor growth inhibition to pathway activity?”* We break this critique down into individual edits and experiments below for ease of discussion:

Comment 3a: *“Can the authors show that in a KRAS wildtype pancreatic cell line, depletion of EGFR3A has no effect?”*

Reply: As requested, we tested and found that the loss of EFR3A or both EFR3A and EFR3B in the KRAS mutation-negative human pancreatic cancer cell line BxPC-3 did not alter RAS signaling, as assessed by measuring phosphorylated (P) P-ERK and P-AKT levels (**Fig S3b**), and had negligible to no effect 2D-transformed growth, as assessed by the colony formation assay (**Fig S3c,d**), and 3D-transformed growth, as assessed by the soft agar assay (**Fig S3a**).

Comment 3b: *“Do the lines they tested exhibit elevated EFR3A expression?”*

Reply: As requested, we note the challenge to this experiment is that we do not have normal matched tissue for the cell lines we analyzed. We did compare the expression of EFR3A to one normal pancreatic biopsy, but found no obvious difference. We do apologize that we were not able to better address this concern.

Comment 3c: *“Can the authors include xenograft study data for the individual depletion of EFR3A and EFR3B.”*

Reply: As requested, we compared the xenograft tumor growth of HPAF-II cells in which EFR3A, EFR3B, or both genes were validated to be inactivated by CRISPR/Cas9-mediated gene inactivation. Consistent with the *in vitro* data, loss of EFR3A or EFR3B nearly abolished tumor growth (**Fig S3e**), while the combination was more potent (**Fig 2j**).

Comment 3d: *“Is there any PD readout from the study, to correlate tumor growth inhibition to pathway activity?”*

Reply: As requested, we tested and found by immunoblot analysis that the loss of EFR3A, EFR3B, or both potently reduced the level of P-ERK and P-AKT in tumor xenografts derived from the KRAS-mutant human pancreatic cancer cell line HPAF-II (**Fig 2k**), consistent with the anti-neoplastic effect of EFR3A and/or B sgRNA due to reduced oncogenic KRAS signaling, for which these cancer cells are addicted to.

Comment 4: *“One of the major issues of the manuscript is evidence for whether the interaction between EFR3A/B with RAS truly exists. The KRAS-EFR3A interaction experiments are weak. First, the interaction is possibly so transient that none of the pull-downs can be done from endogenous protein or with just at least the prey at endogenous levels. The interactions are only seen when both KRAS and EFR3A are ectopically over expressed in cells or when using proximity ligation. With proximity ligation experiments like BioID, the amount of biotin and incubation period can greatly affect the range of interactions. This seems to be an issue in interpreting a few of the results.”* We break this critique down into individual edits and experiments below for ease of discussion:

Overall reply: As requested, we greatly strengthened the case that EFR3A binds to activated KRAS, as we now show that *i*) recombinant EFR3A associates with recombinant KRAS by two different detection methods (co-immunoprecipitation and far westerns (**Fig 3g-j,u-y**), *ii*) this association is dependent upon pre-loading KRAS with non-hydrolyzable GTP γ S but lost when pre-loaded with GDP (**Fig 3g,h**), *iii*) this association occurs with G12V activated, but not the wild-type (unless pre-loaded with GTP γ S) or inactive S17N versions of KRAS (**Fig 3c**), *iv*) the binding of recombinant EFR3A to KRAS^{G12V} maps to a 12 amino acid stretch in the C-terminus of the protein, *v*) highly purified recombinant EFR3A elutes with highly purified recombinant KRAS^{G12V} in a complex, as determined by size exclusion chromatography (**Fig 3q,r**), *vi*) the association of recombinant EFR3A with recombinant KRAS^{G12V} is competed off with the RAS effector BRAF (**Fig 3s**), *vii*) ectopic KRAS with the activating G12V, but not inactivating S17N mutation, co-immunoprecipitates with ectopic EFR3A in cells (**Fig 3c,d**), *viii*) ectopic KRAS^{G12D} co-immunoprecipitates with **endogenous** EFR3A in cells (**Fig 3x,y**), and finally, *ix*) by capturing activated KRAS protein on RBD-conjugated beads from lysates derived from the homozygous KRAS^{G12D}-mutant human pancreatic cancer cell line AsPC-1 followed by immunoblot with an anti-EFR3A antibody, that **endogenous** KRAS associates with **endogenous** EFR3A in cells (**Fig 3z**).

Comment 4a: “Based on the BirA mediated proximity labeling and loss-of-function CRISPR screen, the authors identify EFR3A as a top interactor relative to BRAF. Along the lines, can the authors sparse the dataset based on strong interactors vs. weak interactors.”

Reply: As requested, we now cite these data, which we previously reported in Adhikari & Counter, *Nat Commun* **9**:3646, 2018.

Comment 4b: “Is BRAF used as a relative measure since it’s a strong or a weak interactor of RAS oncoproteins?”

Reply: As requested, we note here that we used BRAF as a standard simply because it is an extremely well-validated RAS effector. BRAF is actually weakly biotinylated, although this could be for many reasons (size of the protein, proximity of the protein to RAS, affinity of the protein for RAS, amount of the protein in the cell, number of exposed lysine residues to facilitate biotinylation, etc.). Our original analysis was to find proteins unique to KRAS signaling, thus we used proximity labeling to simply inform a CRISPR/Cas9 loss-of-function sgRNA library targeting components of the putative interactome. Our first hit (PIP5K1A, see Adhikari & Counter, *Nat Commun* **9**:3646, 2018) was indeed robustly biotinylated by BirA-KRAS (but not by BirA-NRAS or -HRAS) and was required for oncogenic KRAS (but not oncogenic NRAS or HRAS) driven transformed growth. However, mining these two datasets identified *EFR3A* sgRNAs to be consistently the most negatively enriched of all targeted interactome candidates (**Fig 1a**). Thus, even though EFR3A was poorly biotinylated, the average negative enrichment score for all five *EFR3A* sgRNAs in cells transformed by RAS was almost twice that of *BRAF* sgRNAs. For these reasons we pursued EFR3A further. If we have missed the point of this comment, we are happy to revise the text.

Comment 4c: “Given, CRAF and not BRAF is a top synthetic lethal hit in RAS driven cancers, it would be useful to map CRAF or determine EFR3A relative to CRAF”

Reply: As requested, we now compare the biotin labeling and sgRNA enrichment scores of CRAF, BRAF, and EFR3A (**Fig S1d**). Despite CRAF and EFR3A exhibiting almost equivalent biotin labeling, *EFR3A* sgRNAs were more negatively enriched than *CRAF* sgRNAs, which interestingly, were not as negatively enriched as *BRAF* sgRNAs. We note here that others

have found variable enrichment of *BRAF* and *CRAF* sgRNAs in CRISPR/Cas screens (Terrell *et al.*, *Mol Cell* **76**:872, 2019).

Comment 4d: *“Since, BirA screen identifies transient interactions within a 10nM range, do other members of the EFR3 complex which include EFR3B, FAM126A/B, TTC7A/B transiently associated with RAS nanoclustering at the membrane”*

Reply: **As requested**, we performed transmission electron microscopy of GFP-KRAS^{G12V} with mcherry-EFR3B, -FAM126A, and -TTC7A in 293T cells and indeed observed co-clustering of these other components with oncogenic KRAS (**Fig S6**).

Comment 5: *“In Fig 3B the authors pull-down for His₁₀-KRAS-G12V, using a KRAS specific antibody, and probe for GST-EFR3A. In the top panel “CO-IP:EFR3A” the blot shows EFR3A comes down in the KRAS Ab IP, but not in the mock IP with IgG.” We break this critique down into individual edits and experiments below for ease of discussion:*

Comment 5a: *“Is this interaction dependent on KRAS being loaded with GTP or GDP? They compare WT versus mutant KRAS interaction in Figure 3D but never specifically load KRAS with GDP or GTP to show specifically that the interaction is dependent on the active or inactive state of KRAS.”*

Reply: **As requested**, we now note in the figure legend that in **Fig 3e,k** the loading status of KRAS was not confirmed, as noted by the reviewer. Given this, we repeated these experiment with wild-type and G12V active KRAS in the absence of no nucleotide, GDP, and non-hydrolyzable GTP_γS using highly purified recombinant proteins. We confirmed the activation status in all six settings by RBD-affinity capture, which led to the predicable outcome, namely that oncogenic His₁₀-KRAS^{G12V} was more active than His₁₀-KRAS^{WT} in all conditions, and loading with GTP_γS was more active than GDP or buffer with both proteins (**Fig. 3g**). These six versions of His₁₀-KRAS were incubated with highly purified recombinant GST-EFR3A and again His₁₀-KRAS^{G12V} or His₁₀-KRAS^{WT} was immunoprecipitated with a KRAS-specific antibody and immunoblotted with EFR3A- or KRAS-specific antibodies. The results mirrored the activation status of the KRAS proteins, and ranged from GTP_γS loaded His₁₀-KRAS^{G12V} capturing the most GST-EFR3A to buffer or GDP loaded His₁₀-KRAS^{WT} capturing the least (**Fig. 3g,h**). Thus, recombinant GST-EFR3A preferentially co-immunoprecipitates with GTP-bound or oncogenic recombinant His₁₀-KRAS. We also pre-loaded recombinant KRAS^{G12V} with non-hydrolyzable GTP_γS for far western analysis (**Fig 3i,j**).

Comment 5b: *“Can the authors include additional controls here showing that the interaction is specific? Can the authors also conduct a reverse IP? Can they determine the dissociation constant for the interaction? Additional controls with the recombinant proteins specifically would go a long way in convincing the reader that this interaction is specific.”*

Reply: **As requested**, we now *i*) include GST alone as a negative control in **Figs 3i,j,u,v,w** and **S4b** to demonstrate the specificity of the KRAS-EFR3A interaction, *ii*) performed the reverse co-immunoprecipitation in **Fig S4b**, and *iii*) include biophysical size exclusion chromatography using highly purified recombinant protein to validate the association of EFR3A with GTP_γS loaded KRAS^{G12V} (**Fig 3q,r**, also see our reply to concern 1g from reviewer 1 for more details).

Comment 5c: *“Given that the cells used in Fig3B express GST-EFR3A can the authors do the reverse pull-down and IP for EFR3A and probe for KRAS as an alternate way to demonstrate an interaction?”*

Reply: *As requested*, we performed a reverse co-immunoprecipitation assay by immunoprecipitating recombinant GST-EFR3A with a GST antibody and confirmed the presence of recombinant His₁₀-KRAS^{G12V} by a KRAS-specific antibody, validating the direct association between these two proteins (**Fig S4b**).

Comment 5d: *“In Fig 3D increasing amounts of GST-EFR3A is added to the membrane for the dot blot and yet in the bottom panel for the loading control it doesn’t seem like there are increasing amounts of EFR3A. Can the authors account for this discrepancy?”*

Reply: *As requested*, we now include a longer exposure for this figure (now **Fig 3k**) showing increased GST-EFR3A levels.

Comment 5e: *“In the IP using proximity ligation, Fig 3C, EFR3A is biotinylated only in the BirA-KRAS-G12V expressing cell and not KRAS-WT or dominant negative KRAS-S17N suggesting EFR3A associates primarily with active KRAS. What are the RAS-GTP levels in these proximity ligation experiments? Have the authors considered an EGF or growth factor stim experiment in KRAS-WT to see if this increases EFR3A association.”*

Reply: *As requested*, we now *i)* include the RAS-GTP levels of myc-BirA-KRAS mutants when expressed in cells in the new **Fig S4a** and, *ii)* as noted above in our reply to concern 5a, demonstrate a stepwise increase in the association of EFR3A with KRAS by exploiting wild-type and G12V-active KRAS preloaded in buffer, GDP, or GTP γ S, beginning with KRAS^{WT} in the presence of GDP having no measurable GTP-loading and association with EFR3A all the way active KRAS^{G12V} in the presence of GTP γ S having the most GTP-loading and the strongest association with EFR3A (**Fig 3g,h**).

Comment 6: *“The authors argue that KRAS mutant PDAC cells are dependent on EFR3A, likely due to EFR3A’s role in maintaining KRAS localization to the plasma membrane. However, all these effects of EFR3A KO can be explained by EFR3A recruiting PI4KA to the membrane to regulate PI(4)P and PS levels. Thus, loss of EFR3A could broadly impact localization of many membrane bound proteins that includes RAS. As EFR3A can already anchor to the plasma membrane through its N-terminal cysteine rich region the authors have not provided evidence for a requirement of KRAS for EFR3A function.”* We break this critique down into individual edits and experiments below for ease of discussion:

Overall reply: *As requested*, we now make a special note this important interpretation in the discussion. We also note that the association of EFR3A with KRAS is GTP dependent, and that *EFR3A* sgRNA were not negatively enriched in HEK-HT cells lacking oncogenic RAS or in the KRAS wild-type pancreatic cancer cell line BxPC-3. Thus, while we cannot rule out that the loss of EFR3A reduces KRAS signaling by altering the lipid content of the plasma membrane in a manner unrelated to its association with KRAS, the above data support a more targeted effect related to the specific association of active KRAS with EFR3A. In almost all respects, the model outlined by this reviewer is exactly how we envision EFR3A functioning in oncogenic KRAS signaling, the only deviation being that we suggest this effect is dependent upon EFR3A associating with active KRAS.

Comment 6a: *“Given the authors demonstrate that recombinant EFR3A interacts with recombinant mutant KRAS, this suggests that membrane localization may not be required for the interacting regions. Can the authors narrow down potential interaction sequences with truncation mutants of EFR3A as it lacks an RBD domain.”*

Reply: As requested, we performed deletion mapping of EFR3A, and narrowed down residues 700-712 as being required to mediate this direct interaction with KRAS (**Fig 3t-w**). As suggested by the reviewer, this region is not responsible for myristoylation of the protein.

Comment 7: *“Since the authors claim an epistatic relationship with EFA3A/3B and PI4KA, does PI4KA depletion affect ERK signaling? (Fig 5h). The blots are unclear and need to be repeated? Please include a pERK/ERK and pCRAF/total CRAF blots as well. There is also confusion with the writing in the manuscript for figure legends. Figure 3 legend has several errors and do not match with the figure itself. 1. In the figure legend it states that Fig 3D for the dot blot “...probed with increasing amounts of recombinant GST-EFR3A followed by immunoblot with an anti-FLAG antibody.” Which protein contains the FLAG epitope? In Fig 3D the panels indicate anti-EFR3A being used and not FLAG or GST antibodies. 2. In figure legend 3B it’s written as “Levels of oncogenic KRAS or HRAS associating with EFR3A...” but in this case where is HRAS or is this a typo? 3. Fig 3B also states “immunoprecipitating (IP) recombinant His10-KRAS-G12V by virtue of His10-epitope tagged...” doesn’t that mean IP of the His10 tag, thus a Ni pull-down? Yet the figure is showing a KRAS Ab pull-down. We break this critique down into individual edits and experiments below for ease of discussion:*

Comment 7a: *“Since the authors claim an epistatic relationship with EFA3A/3B and PI4KA, does PI4KA depletion affect ERK signaling? (Fig 5h). The blots are unclear and need to be repeated? Please include a pERK/ERK and pCRAF/total CRAF blots as well.”*

Reply: As requested, we tested and now show that phosphorylated (P) P-ERK and P-CRAF^{S338} levels are indeed reduced upon a validated loss of PI4KA in three separate KRAS-mutant human pancreatic cancer cell lines indicative of reduced oncogenic RAS signaling (**Fig 5h**).

Comment 7b: *“In the figure legend it states that Fig 3D for the dot blot “...probed with increasing amounts of recombinant GST-EFR3A followed by immunoblot with an anti-FLAG antibody.” Which protein contains the FLAG epitope? In Fig 3D the panels indicate anti-EFR3A being used and not FLAG or GST antibodies.”*

Reply: As requested, we removed the reference to a FLAG antibody for this panel (which is now **Fig 3k**), as it was a typo, and now clearly indicate that EFR3A was detected with an anti-EFR3A antibody. Thank you for pointing this out.

Comment 7c: *“In figure legend 3B it’s written as “Levels of oncogenic KRAS or HRAS associating with EFR3A...” but in this case where is HRAS or is this a typo? 3. Fig 3B also states “immunoprecipitating (IP) recombinant His10-KRAS-G12V by virtue of His10-epitope tagged...” doesn’t that mean IP of the His10 tag, thus a Ni pull-down? Yet the figure is showing a KRAS Ab pull-down*

Reply: As requested, we revised the figure legend as follows, i) the typo referring to HRAS has been removed, ii) we clarify that that recombinant His₁₀-KRAS^{G12V} was immunoprecipitated with a KRAS-specific antibody and recombinant GST-EFR3A was detected in the immunoprecipitate with an anti-EFR3A specific antibody.

Minor note 1: *“If EFR3B is epistatic to EFR3A is EFR3B upregulated as well in PDACs or other cancers?”*

Reply: As requested, and related to concern 1 of reviewer 2, we extended expression analysis to all the components of the EFR3A signaling complex, and now report an increase in the

expression of *EFR3B* (**Fig1h**), *PI4KA* (**Fig 1i**), *FAM126A* (**Fig 1j**), and *TTC7A* (**Fig 1k**) in pancreatic tumor samples compared to normal tissue. *EFR3A* was not upregulated in four other RAS-mutant cancers examined, a point we bring up in the revised discussion as well.

Minor note 2: *“Fig5H, what about pERK levels?”*

Reply: *As requested*, we now show that P-ERK (as well as P-CRAF and P-AKT) levels are indeed reduced upon the validated loss of *PI4KA* by CRISPR/Cas9 gene inactivation in three independent *KRAS*-mutant human pancreatic cancer cell lines (**Fig 5h**).

Minor note 3: *“Does the PI4Ki inhibitor alone alter RAS association at the membrane? What about effects on MAPK and PI3K signaling?”*

Reply: *As requested*, we tested and now show by transmission electron microscopy that pharmacological inhibition of *PI4KA* by C7 results in a reduction in *KRAS* localization at the plasma membrane and also decreases the plasma membrane phosphatidylserine lipid levels (**Fig 5o**). *As requested*, we tested and now show genetically by the validated loss of *PI4KA* by CRISPR/Cas9 gene inactivation (**Fig 5h**) and pharmacologically with the aforementioned *PI4KA*-specific inhibitor C7 (**Fig 5p**) a reduction in P-CRAF, and/or P-ERK and P-AKT levels in three independent *KRAS*-mutant human pancreatic cancer cell lines.

Minor note 4: *“Fig 5J labels are cut partially off”*

Reply: *As requested*, the labels have been corrected. Thank you for alerting us to this error.

REVIEWERS' COMMENTS

Reviewer #1 (Remarks to the Author):

The authors have done an excellent job in answering all of my questions. They have carried out an excellent set of new experiments that very clearly prove that the EFR-Kras complex forms a direct interaction. This is an important and well performed study, and the manuscript is suitable for publication in Nature Communications

John Burke

Reviewer #2 (Remarks to the Author):

The authors have addressed most of my concerns. A suggestion regarding Fig 3Z: The RasRBD pulldown indeed suggested the interaction between EFR3A and KRAS, yet little to say the specificity to active KRAS (the AsPC 1 has mostly active form of KRAS). Alternatively, the authors can use BxPC3 cells, which express wild type KRAS and respond to EGR stimulation well. It will strengthen their conclusion by showing increased binding of EFR3A to KRAS upon EGF treatment in a dose dependent manner.

Reviewer #3 (Remarks to the Author):

The authors have sufficiently addressed all my concerns particularly with respect to showing a direct interaction with KRAS and EFR3A. I think the experiments with the recombinant protein strengthen the conclusions of the paper.

REVIEWER 2 COMMENT

Comment 1: *“The authors have addressed most of my concerns. A suggestion regarding Fig 3Z: The RasRBD pulldown indeed suggested the interaction between EFR3A and KRAS, yet little to say the specificity to active KRAS (the AsPC 1 has mostly active form of KRAS). Alternatively, the authors can use BxPC3 cells, which express wild type KRAS and respond to EGF stimulation well. It will strengthen their conclusion by showing increased binding of EFR3A to KRAS upon EGF treatment in a dose dependent manner.”*

Reply: As requested, BxPC3 cells were serum starved for 24 hours than treated with EGF and prior to treatment (0 time point) or 5, 10, and 20 minutes after treatment, protein lysates were collected and subjected to affinity purification with an RBD polypeptide followed by immunoblot with an anti-KRAS antibody to detect endogenous GTP-KRAS and an anti-ERF3A antibody to detect co-purified endogenous ERF3A. Original lysates were also immunoblotted with the same two antibodies to measure total levels of each protein. As expected based on the well-established response of mammalian cells to EGF stimulation, GTP-KRAS was not detected before EGF treatment, was detected at 5 and 10 minutes after EGF treatment, then the amount decreased 20 minutes after EGF treatment. The same pattern was seen with EFR3A, namely EFR3A detected in the RBD pull-down was absent at the 0 minute time point, was detected in the 5 and 10 minute time points, then decreased at the 20 minute time point. To ensure reproducibility and rigor, we repeated this experiment a second time, thus adding a second biological replicate. These new data are included as **Fig. S5f**.